 eLIFE

# Mechanistic insights into neurotransmitter release and presynaptic plasticity from the crystal structure of Munc13-1 $C_1C_2BMUN$

Junjie Xu[1,2,3†], Marcial Camacho[4†], Yibin Xu[1,2,3], Victoria Esser[1,2,3], Xiaoxia Liu[1,2,3], Thorsten Trimbuch[4], Yun-Zu Pan[1,2,3], Cong Ma[5,6], Diana R Tomchick[1,2*], Christian Rosenmund[4*], Josep Rizo[1,2,3*]

[1]Department of Biophysics, University of Texas Southwestern Medical Center, Dallas, United States; [2]Department of Biochemistry, University of Texas Southwestern Medical Center, Dallas, United States; [3]Department of Pharmacology, University of Texas Southwestern Medical Center, Dallas, United States; [4]Department of Neurophysiology, NeuroCure Cluster of Excellence, Charité-Universitätsmedizin Berlin, Berlin, Germany; [5]Key Laboratory of Molecular Biophysics of the Ministry of Education, Huazhong University of Science and Technology, Wuhan, China; [6]College of Life Science and Technology, Huazhong University of Science and Technology, Wuhan, China

**\*For correspondence:** diana.
tomchick@utsouthwestern.edu
(DRT); christian.rosenmund@
charite.de (CR); Jose.Rizo-Rey@
UTSouthwestern.edu (JR)

[†]These authors contributed
equally to this work

**Competing interest:** See
page 23

**Reviewing editor:** Reinhard
Jahn, Max Planck Institute for
Biophysical Chemistry, Germany

**Abstract** Munc13–1 acts as a master regulator of neurotransmitter release, mediating docking-priming of synaptic vesicles and diverse presynaptic plasticity processes. It is unclear how the functions of the multiple domains of Munc13–1 are coordinated. The crystal structure of a Munc13–1 fragment including its $C_1$, $C_2B$ and MUN domains ($C_1C_2BMUN$) reveals a 19.5 nm-long multi-helical structure with the $C_1$ and $C_2B$ domains packed at one end. The similar orientations of the respective diacyglycerol- and $Ca^{2+}$-binding sites of the $C_1$ and $C_2B$ domains suggest that the two domains cooperate in plasma-membrane binding and that activation of Munc13–1 by $Ca^{2+}$ and diacylglycerol during short-term presynaptic plasticity are closely interrelated. Electrophysiological experiments in mouse neurons support the functional importance of the domain interfaces observed in $C_1C_2BMUN$. The structure imposes key constraints for models of neurotransmitter release and suggests that Munc13–1 bridges the vesicle and plasma membranes from the periphery of the membrane-membrane interface.

## Introduction

The release of neurotransmitters by synaptic vesicle exocytosis is critical for neuronal communication. This exquisitely regulated process involves several steps, including tethering of synaptic vesicles to specialized sites of the presynaptic plasma membrane called active zones, a priming step(s) that leaves the vesicles ready to release, and $Ca^{2+}$-triggered membrane fusion (*Südhof, 2013*). Each of these steps can be modulated in a variety of presynaptic plasticity processes that underlie multiple forms of information processing in the brain (*Regehr, 2012*). The protein machinery that controls release (*Rizo and Xu, 2015*; *Jahn and Fasshauer, 2012*; *Südhof and Rothman, 2009*) includes the neuronal soluble N-ethylmaleimide-sensitive factor attachment protein receptors (SNAREs) synaptobrevin, syntaxin-1 and SNAP-25, which play a key role in membrane fusion by forming a tight four-

**eLife digest** The human brain contains billions of cells called neurons that communicate with each other using molecules called neurotransmitters. An electrical signal in one neuron triggers the release of neurotransmitters from the cell, which then activate or inhibit electrical signals in neighboring neurons. Inside the cell, neurotransmitters are stored in small bubble-like structures called synaptic vesicles. The vesicles fuse with the membrane that surrounds the cell to release the neurotransmitters. This process must be tightly controlled to ensure that neurotransmitters are released rapidly and at the right time.

A protein called Munc13 is a key component of the machinery that regulates the fusion of synaptic vesicles. It helps the synaptic vesicle to dock onto the cell membrane and get ready for fusion. Munc13 is a large protein and contains several different regions, including three domains called $C_1$, $C_2B$ and MUN. These three domains control the release of neurotransmitters, but how they do so is poorly understood.

Xu, Camacho et al. used a technique called X-ray crystallography to analyse the three-dimensional shape of the part of Munc13 that contains the three domains. The experiments reveal that the MUN domain forms a long rod-like shape with the $C_1$ and $C_2B$ domains packed at one end. Several mutations that reduce the ability of the domains to interact with each other altered the release of neurotransmitters from mouse neurons to different extents.

These findings suggest that the overall architecture of the region containing the $C_1$, $C_2B$ and MUN domains is important for the normal activity of Munc13. The structure revealed by Xu, Camacho et al. sets a framework for understanding how Munc13 controls neurotransmitter release, and thus mediates diverse forms of information processing in the brain.

helix bundle (the SNARE complex) that brings the vesicle and plasma membranes into close proximity (*Söllner et al., 1993*; *Hanson et al., 1997*; *Poirier et al., 1998*; *Sutton et al., 1998*). This complex is disassembled by N-ethylmaleimide-sensitive factor (NSF) and soluble NSF attachment proteins (SNAPs; no relation to SNAP-25) (*Söllner et al., 1993*) to recycle the SNAREs for another round of fusion (*Mayer et al., 1996*; *Banerjee et al., 1996*). The Sec1/Munc18 protein Munc18–1 and the large (200 kDa) active zone proteins called Munc13s are also crucial for release. Munc18–1 binds to a self-inhibited 'closed' conformation of syntaxin-1 (*Dulubova et al., 1999*; *Misura et al., 2000*) and orchestrates SNARE complex assembly in an NSF-SNAP-resistant manner together with Munc13, which helps to open syntaxin-1 (*Richmond et al., 2001*; *Ma et al., 2011, 2013*).

Because of their multiple functions, Munc13s have emerged as particularly central regulators of neurotransmitter release that link the core membrane fusion apparatus to diverse forms of presynaptic plasticity through their multidomain architecture (illustrated in *Figure 1A* for Munc13–1, the most abundant mammalian isoform). Thus, neurotransmitter release is completely abrogated in the absence of Munc13s (*Augustin et al., 1999*; *Richmond et al., 1999*; *Aravamudan et al., 1999*; *Varoqueaux et al., 2002*). This phenotype most likely arises because Munc13s play key roles in docking and priming (*Varoqueaux et al., 2002*; *Weimer et al., 2006*; *Hammarlund et al., 2007*; *Imig et al., 2014*) that are associated at least in part to their activity in opening syntaxin-1 and thus stimulating SNARE complex formation through their MUN domain (*Richmond et al., 2001*; *Basu et al., 2005*; *Yang et al., 2015*). However, the $C_1$-$C_2B$ region and the C-terminal $C_2C$ domain are also critical for release, which may arise because these domains help bridge the vesicle and plasma membranes (*Liu et al., 2016*). There is also evidence for a role of Munc13s in events downstream of priming [e.g. (*Hammarlund et al., 2007*; *Shin et al., 2010*; *Liu et al., 2016*). Moreover, Munc13s are involved in multiple presynaptic plasticity processes, including isoform-specific depression and augmentation (*Rosenmund et al., 2002*), diacylglycerol (DAG)-phorbol ester-dependent potentiation of release via the $C_1$ domain (*Rhee et al., 2002*) and $Ca^{2+}$-dependent short-term plasticity through the $C_2B$ domain and a calmodulin-binding region (*Shin et al., 2010*; *Junge et al., 2004*). Binding of the Munc13–1 $C_2A$ domain to RIMs also provides a connection to RIM-dependent forms of short- and long-term presynaptic plasticity (*Betz et al., 2001*; *Dulubova et al., 2005*). The crucial physiological importance of these modulatory processes was emphatically illustrated by the

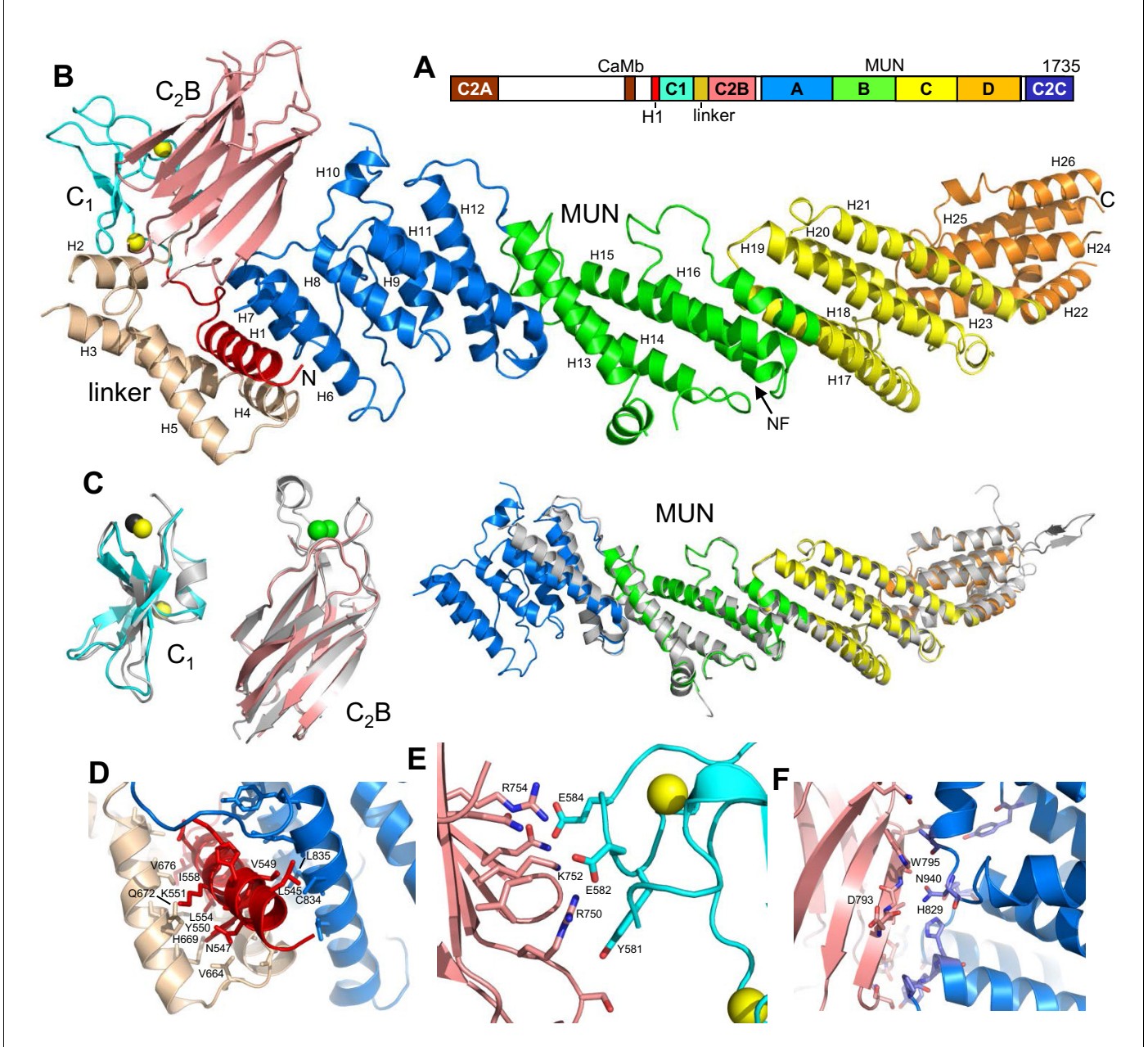

**Figure 1.** Crystal structure of Munc13–1 $C_1C_2BMUN$. (**A**) Domain diagram of rat Munc13–1, with A-D corresponding to the four subdomains of the MUN domain. CaMb = calmodulin-binding sequence. (**B**) Ribbon representation of the structure of Munc13–1 $C_1C_2BMUN$ color-coded as in the domain diagram. Helices are numbered and labeled. The position of the NF sequence involved in opening syntaxin-1 is indicated. The $Zn^{2+}$ ions bound to the $C_1$ domain are shown as yellow spheres. (**C**) Superimposition of the structures of the Munc13–1 isolated $C_1$ domain (PDB code 1Y8F), $Ca^{2+}$-bound $C_2B$ domain (PDB code 3 KWU) and our refined structure of the almost complete MUN domain with the structures of these domains in the crystal structure of $C_1C_2BMUN$ [r.m.s.d. between equivalent $C\alpha$ atoms are 1.15 (45 $C\alpha$ atoms), 0.35 (85 $C\alpha$ atoms) and 0.96 (465 $C\alpha$ atoms), respectively]. $Ca^{2+}$ ions of the $C_2B$ domain structure are shown as green spheres. D-F. Interfaces of helix H1 with the linker helices and with the MUN domain (**D**), of the $C_1$ domain with the $C_2B$ domain (**E**) and of the $C_2B$ domain with the MUN domain (**F**). Selected sides chains in the interfaces, including those that were mutated, are indicated.

The following figure supplements are available for figure 1:

**Figure supplement 1.** $Zn^{2+}$ ions in the $C_1$ domain of $C_1C_2BMUN$.

**Figure supplement 2.** Comparison of chains A (green) and B (red) of C1C2BMUN.

*Figure 1 continued on next page*

*Figure 1 continued*

**Figure supplement 3.** Lattice contacts for C1C2BMUN domains, C2 symmetry.

**Figure supplement 4.** Plot of mean atomic displacement parameters (*B*-factors) versus residue number for the C1C2BMUN fragment.

**Figure supplement 5.** Omit maps for domain interfaces where mutations were made.

finding that knock-in mice bearing a point mutation in the Munc13–1 $C_1$ domain that disrupts phorbol-ester-dependent potentiation of release die 2–3 hr after birth even though the mutation causes no impairment of evoked release (*Rhee et al., 2002*).

Despite these advances and the availability of three-dimensional structures for most of the Munc13 domains except $C_2C$ and part of the MUN domain (*Shen et al., 2005*; *Dulubova et al., 2005*; *Rodríguez-Castañeda et al., 2010*; *Shin et al., 2010*; *Li et al., 2011*; *Yang et al., 2015*), it is still unclear how the functions of these domains are related and coordinated, in part because no structure of a fragment containing multiple Munc13 domains has been described. Here, we report the crystal structure of a Munc13–1 fragment spanning its $C_1$, $C_2B$ and MUN domains ($C_1C_2BMUN$), revealing a long, 195 Å rod formed by 26 α-helices that packs at one end against the $C_1$ and $C_2B$ domains. The DAG-binding region of the $C_1$ domain and the $Ca^{2+}$-binding region of the $C_2B$ domain are near each other and point in the same direction, which is expected to facilitate cooperation between the two domains in membrane binding and thus enable synergy between the effects of DAG and $Ca^{2+}$ in neurotransmitter release during repetitive stimulation. Electrophysiological experiments show that mutations designed to disrupt interfaces between different domains of $C_1C_2BMUN$ impair evoked release and vesicle priming to different extents, and also have differential effects on $Ca^{2+}$-dependent short-term plasticity as well as phorbol ester-induced potentiation. These results suggest that the highly elongated nature of $C_1C_2BMUN$ and the relative disposition of its domains in our crystal structure are critical for the normal functions of Munc13–1 in neurotransmitter release and presynaptic plasticity, placing important structural constraints on possible models for the mechanisms of neurotransmitter release and presynaptic plasticity.

## Results

### Crystal structure of Munc13–1 $C_1C_2BMUN$

The work presented herein culminates 12 years of efforts dedicated to determine the three-dimensional structures of fragments encompassing part of or the entire highly conserved C-terminal region of Munc13–1, which includes the $C_1$, $C_2B$, MUN and $C_2C$ domains (*Figure 1A*). While crystals were obtained for several of the fragments that we prepared, they tended to diffract poorly. Key for the success in obtaining crystals of $C_1C_2BMUN$ of sufficient quality to diffract in the 3–3.5 Å range were the choice of N- and C-termini (residues 529 and 1531, respectively), as well as the removal of residues 1408–1452, which correspond to a long loop within the MUN domain that is poorly conserved and is subject to alternative splicing (*Brose et al., 1995*). Removal of this loop generally increases the solubility of C-terminal Munc13–1 fragments (*Ma et al., 2011*; *Li et al., 2011*). Even with the best diffraction data obtained with $C_1C_2BMUN$, structure determination was hindered by low resolution, significant anisotropy, non-isomorphism and an inability to obtain selenomethionyl-derivatized protein. These problems were overcome by the use of single wavelength anomalous dispersion phases obtained from a dataset collected at the tantalum LIII edge on native crystals soaked with a tantalum bromide cluster, coupled with molecular replacement phases obtained from the previously determined structures of the $C_1$ domain (*Shen et al., 2005*), the $C_2B$ domain (*Shin et al., 2010*) and the nearly complete MUN domain of Munc13–1. For this purpose, it was necessary to first re-refine the structure of the nearly complete MUN domain (*Yang et al., 2015*) using the deposited structure factors (PDB code 4Y21) to reduce the level of side chain outliers and of steric clashes, as well as to correct the sequence numbering (see Materials and methods). Placement of the $C_1$ domains in the cell of the $C_1C_2BMUN$ crystals (*Figure 1—figure supplement 1*) was verified by an anomalous difference map calculated from data collected on native crystals at the zinc K-edge energy. Data

collection and refinement statistics for the final structure of $C_1C_2BMUN$, as well as for the re-refined structure of the nearly complete MUN domain, are described in *Table 1*.

The structures of the two molecules of $C_1C_2BMUN$ present in the asymmetric unit of its crystals are very similar (*Figure 1—figure supplement 2*). The structure of $C_1C_2BMUN$ can be viewed as a highly elongated rod spanning ca. 195 Å and is formed mostly of α-helical bundles, with a total of 26 helices; the $C_1$ and $C_2B$ domains pack at the N-terminal end of the rod (*Figure 1B*). Much of the long rod is formed by the MUN domain, which is homologous to subunits of complexes that mediate tethering in diverse membrane compartments (*Pei et al., 2009*; *Yu and Hughson, 2010*) and was previously shown to be formed by four subdomains of about five helices each (named A-D and colored in blue, green, yellow and orange on *Figure 1*) (*Li et al., 2011*; *Yang et al., 2015*). Comparison of our $C_1C_2BMUN$ structure with the crystal structure of the almost complete MUN domain (with the N-terminus at residue 933), which was described in *Yang et al. (2015)* and we re-refined, shows that the MUN domain is very similar in both structures (*Figure 1C*), as expected because this structure was used in crystallographic phase determination by the molecular replacement method. However, the structure of $C_1C_2BMUN$ shows that the MUN domain actually starts at residue 828 and contains five additional helices (helices H6-H10), resulting in a total of seven helices for subdomain A (*Figure 1C*). In addition, there are five more helices preceding the MUN domain (helices H1-H5; *Figure 1B*). Four of these helices correspond to the linker sequence between the $C_1$ and $C_2B$ domains, whereas one helix (H1) is spanned by a sequence preceding the $C_1$ domain (colored in red).

These observations nicely explain a few findings made during our crystallization efforts. For instance, Munc13–1 fragments starting right at the beginning of the $C_1$ domain were unstable and we were not able to express fragments containing only the $C_1$ domain, the $C_2B$ domain and the linker between them in soluble form. As the structure of $C_1C_2BMUN$ now reveals, there are extensive contacts between helix H1, the four helices of the linker, the MUN domain, the $C_1$ domain and the $C_2B$ domain (*Figure 1B,D–F*), and hence it is not surprising that elimination of some of these interfaces impairs proper folding. We also note that the structures of the $C_1$ and $C_2B$ domains in $C_1C_2BMUN$ are similar to those determined for the isolated domains (*Shen et al., 2005*; *Shin et al., 2010*), although the $Ca^{2+}$-binding loops are not visible in the $C_2B$ domain of $C_1C_2BMUN$ (*Figure 1C*). This correlates with the previous finding that these loops were observable in the crystal structure of the $C_2B$ domain bound to $Ca^{2+}$ but not in its crystal structure without $Ca^{2+}$ (*Shin et al., 2010*), as our $C_1C_2BMUN$ structure was determined in the absence of $Ca^{2+}$.

The architecture of $C_1C_2BMUN$ has important implications to understand the functions of Munc13–1 in docking and priming, as well as the regulatory roles of its various domains. First, the elongated structure and the overall packing of the different domains at the N-terminal end of the structure suggest that $C_1C_2BMUN$ may function as a rigid or semi-rigid unit that bridges the synaptic vesicle and plasma membranes. Note in this context that the similarity in the structures of the two molecules from the asymmetric unit of the $C_1C_2BMUN$ crystals (*Figure 1—figure supplement 2*) and between these structures and that of the nearly complete MUN domain (*Figure 1C*) suggest that the overall architecture of $C_1C_2BMUN$ has limited flexibility. Second, a model of the structure of $C_1C_2BMUN$ incorporating the $Ca^{2+}$-binding loops of the $C_2B$ domain, which mediate its interactions with $PIP_2$-containing membranes in a $Ca^{2+}$-dependent manner (*Shin et al., 2010*), shows that these loops are proximal to the DAG-phorbol ester-binding region of the $C_1$ domain (*Figure 2A,B*). This arrangement is expected to promote cooperation between the $C_1$ and $C_2B$ domains in membrane binding, thus suggesting a natural mechanism for synergy between increases in DAG and intracellular $Ca^{2+}$ concentrations to enhance release probability upon repetitive stimulation. However, it is noteworthy that there are abundant basic residues in the $C_1$-$C_2B$ region (*Figure 2A*) that could potentially mediate membrane binding in alternative orientations in the absence of DAG and $Ca^{2+}$-bound to the $C_2B$ domain (e.g. *Figure 2C*). These observations suggest that increased DAG and intracellular $Ca^{2+}$ concentrations may alter release probability by promoting a different orientation of the Munc13–1 C-terminal region that is more efficient in promoting priming and/or fusion (see Discussion and models described therein). Third, the finding that helix H1 is packed against the MUN domain, with the N-terminus pointing toward the center of the long rod (*Figure 1B*), suggests that N-terminal sequences of Munc13–1 not included in this structure could perform their regulatory functions by influencing MUN domain activity. For instance, the calmodulin-binding region of Munc13–1 might inhibit release by binding to the middle of the MUN domain, where an NF

**Table 1.** Data collection and refinement statistics.

**Data collection**

| Crystal | Ta LIII-edge peak[*] | Zn K-edge peak[*] | MUN domain | Native |
|---|---|---|---|---|
| Space group | C2 | C2 | P2₁2₁2 | C2 |
| Cell constants (Å,°) | 171.70 Å, 82.93 Å, 201.59 Å, 90.0°, 115.32°, 90.0° | 174.72 Å, 84.55 Å, 202.10 Å, 90.0°, 115.11°, 90.0° | 114.1 Å, 270.9 Å, 47.7 Å, 90.0°, 90.0°, 90.0° | 176.13 Å, 86.34 Å, 202.13 Å, 90.0°, 115.54°, 90.0° |
| Wavelength (Å) | 1.25489 | 1.28218 | 0.979 | 0.97931 |
| Resolution range (Å) | 42.73–4.50 (4.64–4.50) | 49.81–4.00 (4.07–4.00) | 39.02–2.90 (2.97–2.90) | 45.60–3.35 (3.41–3.35) |
| Unique reflections | 12,839 (620) | 21,415 (961) | 33,802 (2,770) | 37,636 (1,363) |
| Multiplicity | 3.6 (2.8) | 3.8 (2.5) | 5.4 (5.1) | 3.9 (3.3) |
| Data completeness (%) | 93.1 (64.4) | 94.3 (86.4) | 99.3 (98.7) | 93.9 (68.9) |
| $R_{merge}$ (%)[†] | 3.1 (40.0) | 10.1 (100.0) | 6.9 (63.4) | 5.6 (69.2) |
| $R_{pim}$ (%)[‡] | 1.8 (25.7) | 5.7 (76.5) | NA | 3.1 (38.8) |
| $CC_{1/2}$ (outermost resolution shell) | 0.974 | 0.490 | NA | 0.809 |
| I/σ(I) | 25.0 (0.8) | 30.0 (2.1) | 27.3 (1.95) | 20.7 (1.3) |
| Wilson B-value (Å²) | | | 85.4 | 104.7 |
| Wilson B-value, sharpened (Å²)[§] | 32.6 | 209.2 | NA | 49.6 |

**Refinement statistics**

| | | | | |
|---|---|---|---|---|
| Resolution range (Å) | | | 39.02–2.90 (2.98–2.90) | 45.60–3.35 (3.46–3.35) |
| No. of reflections $R_{work}$/$R_{free}$ | | | 33,801/1,712 (2,634/136) | 29,935/1,490 (752/34) |
| Data completeness (%) | | | 99.2 (98.0) | 75.3 (22.0) |
| Atoms (non-H protein/ $Zn^{2+}$/$Cl^−$) | | | 4286/NA/NA | 13,597/4/2 |
| $R_{work}$ (%) | | | 22.8 (34.5) | 25.4 (32.7) |
| $R_{free}$ (%) | | | 25.3 (35.2) | 29.0 (44.5) |
| R.m.s.d. bond length (Å) | | | 0.002 | 0.003 |
| R.m.s.d. bond angle (°) | | | 0.499 | 0.610 |
| Mean B-value (Å²) (chain A/chain B/ $Zn^{2+}$/$Cl^−$) | | | 101.6/NA/NA | 78.8/53.2/44.4/8.2 |
| Ramachandran plot (%) (favored/additional/ disallowed)[#] | | | 95.3/3.9/0.8 | 92.1/6.7/1.2 |
| Clashscore/Overall score[#] | | | 2.57/1.37 | 3.51/1.62 |
| Maximum likelihood coordinate error | | | 0.38 | 0.41 |
| Missing residues | | | 933–941, 1041–1049, 1524–1531 | A: 529–540, 704–707, 759–773, 801–807, 821–823, 923–928, 1038–1052, 1191–1196, 1342–1352, 1404–1469, 1518–1531. B: 529–541, 626-630, 703–708, 743–745, 759–774, 802–806, 820–824, 925–929, 1038–1050, 1338–1352, 1405–1467, 1516–1531. |

Data for the outermost shell are given in parentheses.

*Bijvoet-pairs were kept separate for data processing.

$^{†}R_{merge} = 100 \sum_h \sum_i |I_{h,i} - \langle I_h \rangle| / \sum_h \sum_i \langle I_{h,i} \rangle$, where the outer sum (h) is over the unique reflections and the inner sum (i) is over the set of independent observations of each unique reflection.

$^{‡}R_{pim} = 100 \sum_h \sum_i [1/(n_h - 1)^{1/2}]|I_{h,i} - \langle I_h \rangle|/ \sum_h \sum_i \langle I_{h,i} \rangle$, where $n_h$ is the number of observations of reflections h.

$^{§}$B-factor sharpening was performed in the autocorrection mode of HKL-3000 (**Borek et al., 2013**).

$^{#}$As defined by the validation suite MolProbity (**Chen et al., 2010**).

sequence (**Figure 1B**) is known to be critical for opening syntaxin-1 (**Yang et al., 2015**), and binding of $Ca^{2+}$-calmodulin to this region may enhance release by relieving this inhibition. Interactions of homodimerized Munc13–1 $C_2A$ domain (**Lu et al., 2006**) with the MUN domain might also underlie the inhibition caused by homodimerization, which is relieved by RIM binding (**Deng et al., 2011**).

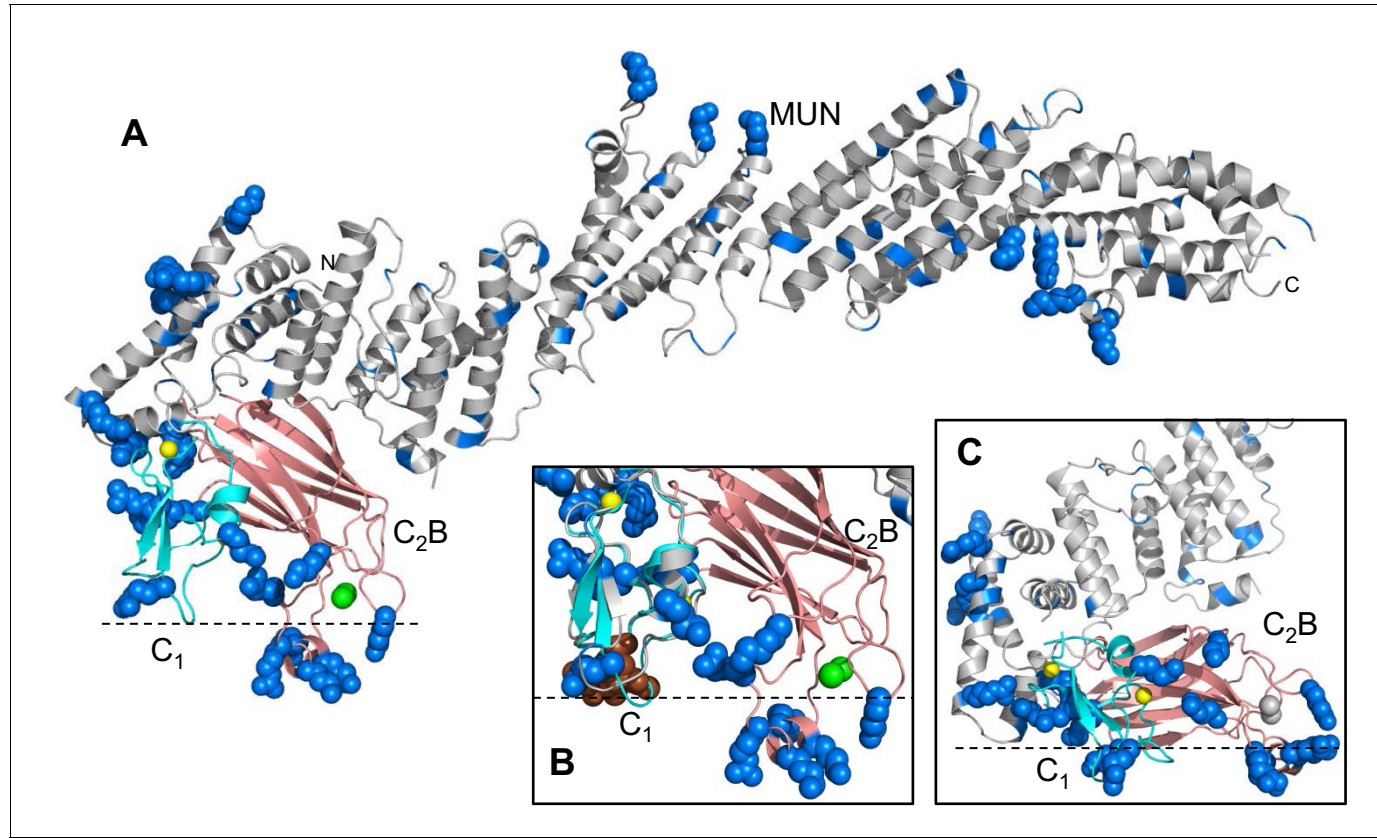

**Figure 2.** Distribution of basic residues clusters in $C_1C_2BMUN$. (**A**) Ribbon diagram of $C_1C_2BMUN$ where the $C_1$ and $C_2B$ domains were replaced with the structures of the isolated $C_1$ domain (cyan; PDB code 1Y8F) and the isolated $Ca^{2+}$-bound $C_2B$ domain (salmon; PDB code 3 KWU) (**Shin et al., 2010**) to include basic residues that are not observed in the structure of $C_1C_2BMUN$, likely because they are disordered. Arginine and lysine side chains are shown as blue spheres. The $Zn^{2+}$ ions in the $C_1$ domain are shown as yellow spheres and two $Ca^{2+}$ ions bound to the $C_2B$ domain are shown as green spheres. Note that the two $Ca^{2+}$ ions are not observed in the $C_1C_2BMUN$ structure, which was crystallized in the absence of $Ca^{2+}$. (**B**) Close up of the structure of $C_1C_2BMUN$ as shown in A, but including a phorbol ester drawn as brown spheres. The position of the phorbol ester is based on superimposing the $C_1$ domain of $C_1C_2BMUN$ with the phorbol-ester bound structure of the PKC-δ $C_1B$ domain (PDB code 1PTR) (**Zhang et al., 1995**). The dashed line indicates the expected approximate location of the plasma membrane bound to the $C_1$ domain through DAG and to the $Ca^{2+}$-bound $C_2B$ domain through $PIP_2$. (**C**) Close up of the same region of $C_1C_2BMUN$ but shown in another orientation with a region containing multiple basic side chains (most from the $C_1$ and $C_2B$ domains) at the bottom. The $Ca^{2+}$-binding sites of the $C_2B$ domain are shown in gray to symbolize that the sites are not occupied. The dashed line indicates a potential localization of the plasma membrane resulting from binding of this basic face to PS, which is sufficient for binding of $C_1C_2BMUN$ to liposomes even in the absence of $Ca^{2+}$, DAG and $PIP_2$ (**Liu et al., 2016**).

## Functional consequences caused by disruption of $C_1C_2$BMUN domain interfaces

A key question that arises from the crystal structure of $C_1C_2$BMUN is whether the arrangement of the different domains at the N-terminal end of the structure is caused by crystal packing or reflects the native structure of Munc13–1 in neurons, and hence is important for its functions. To address this question, we designed three mutations to strongly disrupt domain interfaces observed in the $C_1C_2$BMUN structure: (i) a V549E,L554E double mutation designed to disrupt the hydrophobic packing of helix H1 against the linker region and against the MUN domain (*Figure 1D*); (ii) an R750E, K752E double mutation to disrupt salt bridges formed between the $C_1$ and $C_2$B domains (*Figure 1E*); and (iii) an N940W mutation designed to perturb the interface between the $C_2$B and MUN domains (*Figure 1F*). We tested the physiological consequences of these mutations by performing rescue experiments in autaptic hippocampal neuronal cultures from Munc13-1/2 double KO mice (*Varoqueaux et al., 2002*) that express WT or mutant Munc13–1s through a lentiviral vector. To dissect which aspects of neurotransmitter release and its regulation might be affected by the mutations, we evaluated spontaneous release, $Ca^{2+}$-dependent release triggered by a single action potential (AP), vesicle priming, high-frequency stimulation and PDBu-induced facilitation. Western blot analyses showed that the WT and mutant proteins were detectable at comparable levels (*Figure 3—figure supplement 1*). Although we cannot completely rule out the possibility that the differences observed in our electrophysiological measurements arise in part from distinct expression levels, this possibility is unlikely because no significant changes in synaptic properties are observed in heterozygous Munc13–1 (+/-) neurons and WT neurons mildly overexpressing WT munc13–1 in a Munc13–2 KO background (MC and CR, unpublished results).

We first assessed the effects on spontaneous release measuring the frequency and the amplitude of miniature excitatory postsynaptic currents (mEPSCs). Only the V549E,L554E mutation in the N-terminal H1 helix showed an increase in mEPSC frequency, whereas the mEPSC charge and amplitude were unchanged (*Figure 3*). We next recorded excitatory postsynaptic currents (EPSCs) induced by a single AP to characterize evoked release. Analysis of the EPSC amplitudes revealed that the R750E,K752E mutation in the $C_1$-$C_2$B interface and the N940W mutation in the C2B-MUN interface impaired $Ca^{2+}$-triggered release (*Figure 4A,B*). However, expression of the V549E,L554E mutant where helix H1 is perturbed fully rescued evoked release.

To dissect putative changes in the parameters that underlie evoked release, we first quantified the size of the readily releasable pool (RRP) by measuring the responses induced by hypertonic solution (*Rosenmund and Stevens, 1996*). We found that the N940W mutation in the $C_2$B-MUN interface caused a decrease in RRP size of close to 40%, similar to the change seen in the evoked response. The H1 helix mutation V549E,L554E decreased the RRP size by nearly 50% (*Figure 4C,D*), which contrasts with the lack of an effect of this mutation on evoked release. Furthermore, the disruption of the $C_1$-$C_2$B interface by the R750E,K752E mutation led to normal vesicle priming despite reducing the evoked response. These experiments demonstrate that all three mutations impact Munc13–1 function but, interestingly, each mutation displays distinct effects on evoked release and vesicle priming. This finding strongly suggests that the domain interfaces differentially regulate the efficiency of the vesicle fusion process, which may lead to differences in vesicle release probability.

We calculated the vesicular release probability (Pvr) by dividing the EPSC charge by the charge of the RRP. The V549E,L554E mutation in the H1 helix increased the vesicular release probability significantly, as expected, while the small differences in Pvr observed for the R750E,K752E and N940W mutants with respect to WT were not statistically significant (*Figure 4E*). The impact on release probability in these mutants was further corroborated by analyzing the degree of facilitation or depression quantified by the paired pulse ratio of two consecutive AP-induced EPSC amplitudes (*Figure 4F,G*). Compared to WT, the V549E,L554E mutant showed more depression, in correlation with our finding of enhanced release probability, while the other two mutants showed similar paired-pulse behavior to WT rescues, in agreement with the conclusion that these mutations did not disrupt basal release probability substantially.

In general, vesicular release probability is also a major determinant for release during sustained AP trains. Initial high vesicular release probability predicts more depression of synaptic responses due to more prominent depletion of primed vesicles. However, structure function studies on Munc13 have identified changes in sustained release that are not explained by the initial release

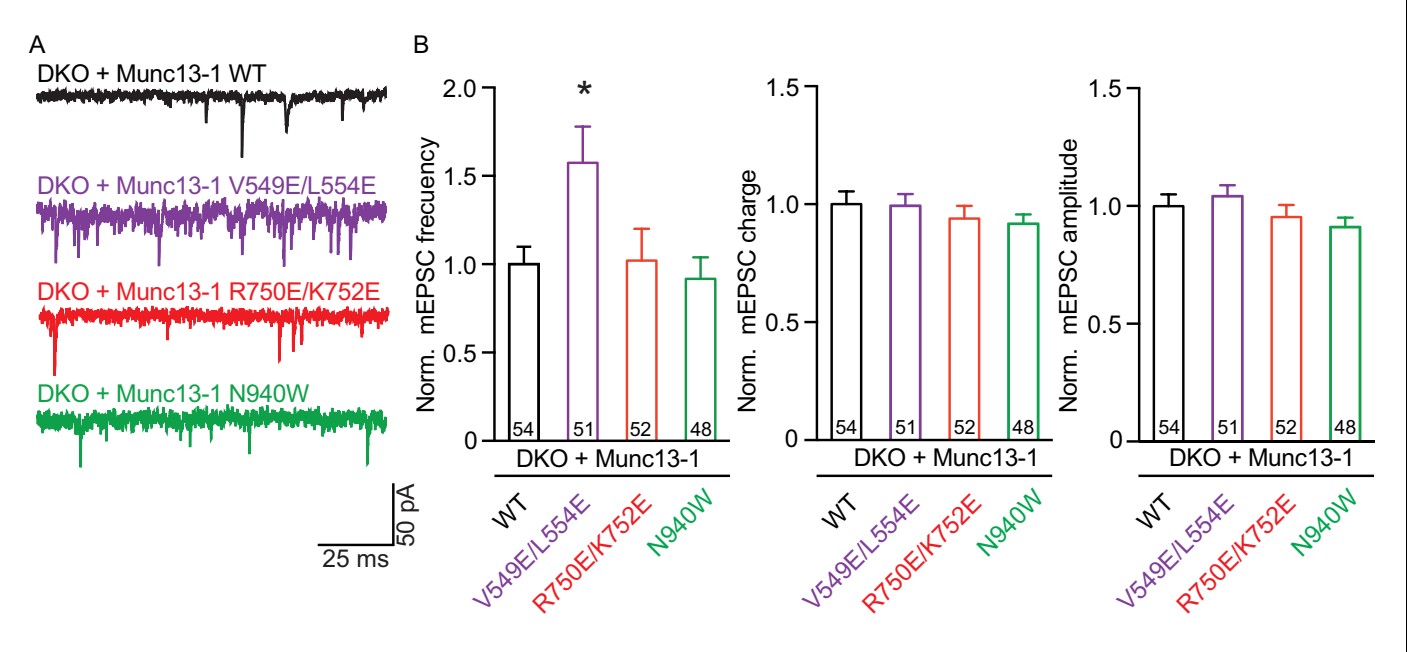

**Figure 3.** Effect on spontaneous release in synapses expressing Munc13–1 mutants that disrupt the $C_1C_2$BMUN domain interfaces. (A) Representatives traces of spontaneous release on synapses from Munc13-1/2 DKO rescued with the respective Munc13–1 WT and mutants indicated above. (B) Plots of mEPSC frequency, charge and amplitudes of Munc13–1 mutants normalized to corresponding Munc13–1 WT. Numbers in plots are *n* values for each group. Error bars represent SEM. Significance and p values were determined by comparison with the corresponding WT using the unpaired Student's *t* test: Mann-Whitney. *p<0.05.

The following figure supplement is available for figure 3:

**Figure supplement 1.** Detection of protein expression from Munc13-1/2 DKO hippocampal neurons.

probability, and this has been interpreted as Munc13 playing an additional role in synaptic augmentation (*Rosenmund et al., 2002*) and in modifying vesicle replenishment through activity-dependent vesicle re-priming (*Shin et al., 2010*). To test how disruption of the interfaces in Munc13–1 can affect the activity-dependent vesicle re-priming, we monitored synaptic responses during 10 Hz AP trains (*Figure 5*). We first defined the function of steady-state depression versus release probability in WT neurons by applying AP trains at various external $Ca^{2+}$ concentrations, and found as expected a robust correlation between release probability and steady state depression (*Figure 5B*). Interestingly, the V549E,L554E mutation in the H1 helix and the R750E,K752E mutation in the $C_1$-$C_2$B interface caused more pronounced depression of sustained release than expected from the initial release probability, while disrupting the $C_2$B-MUN interface with the N940W mutation caused less depression than expected from the initial release probability (*Figure 5B,C*). These results suggest that each one of the interfaces disrupted by the mutations plays a role in activity-dependent vesicle re-priming, and reinforce the notion that Munc13 exerts multiple functions in shaping release properties at synapses.

The regulatory function of Munc13 can also be probed by applying phorbol esters that act as exogenous agonists of the $C_1$ domain and increase the vesicle release probability. To test how the mutations in the domain interfaces of Munc13–1 alter the coupling of phorbol-ester binding to changes in release probability, we examined the effects of acute application of 1 μM PDBu, which causes a 60% increase in the evoked postsynaptic responses in WT neurons (*Figure 6*). The N940W mutation in the $C_2$B-MUN interface showed more pronounced potentiation than WT Munc13–1 (*Figure 6B*), which correlates with the reduction in steady-state depression during 10 Hz AP trains observed for this mutant (*Figure 5C*) and indicates an easier transition for Munc13–1 to a potentiated state. Conversely, the V549E,L554E in helix H1 led to reduced potentiation by PDBu compared to WT, which likely arises because of the initial high vesicular release probability observed for this

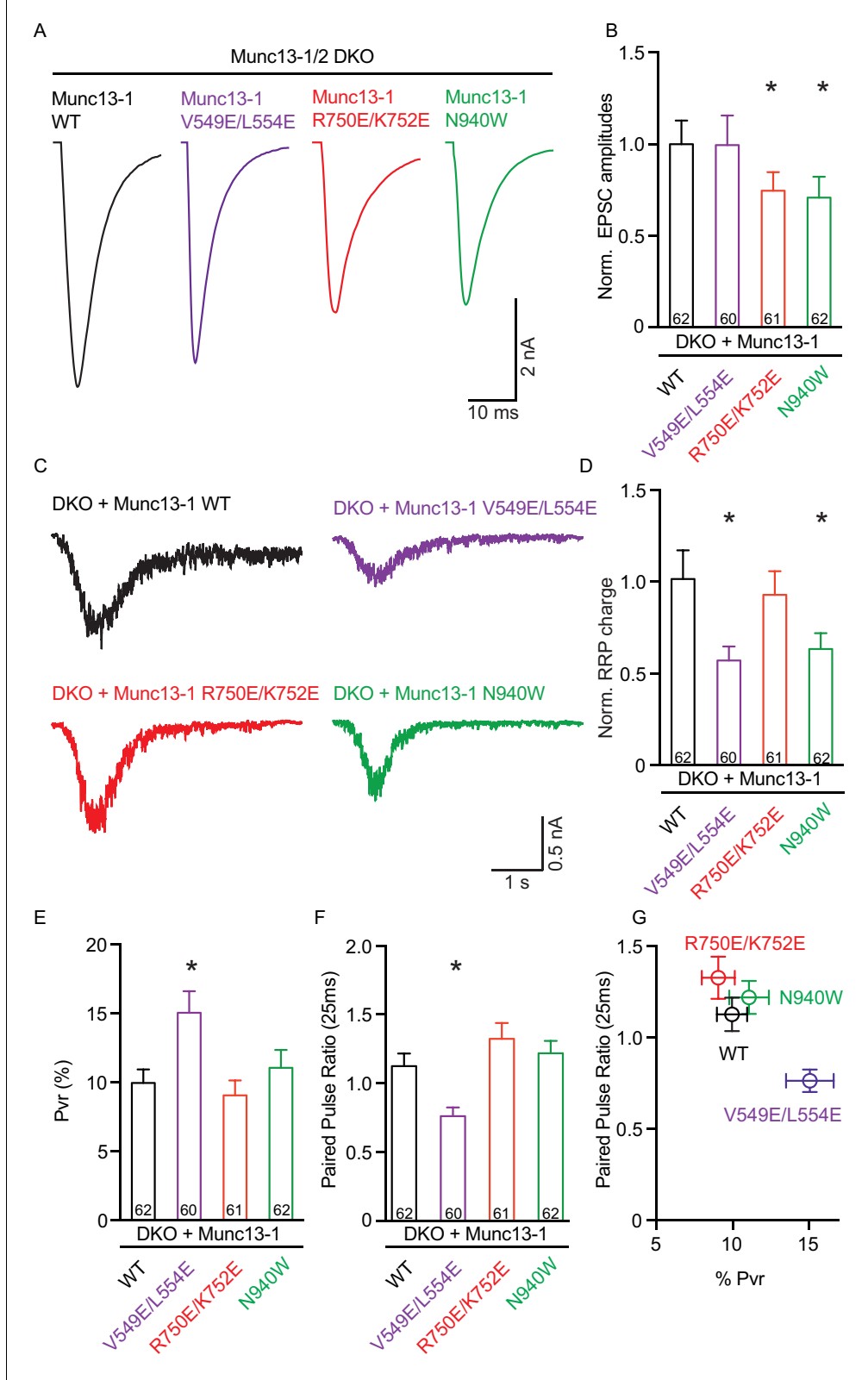

**Figure 4.** Single action potential evoked synaptic transmission, ready releasable pool and release probability consequences by the disruption of the $C_1C_2B$MUN domain interfaces. (**A**) Representatives traces of AP-evoked EPSC amplitudes recorded form Munc13-1/2 DKO and DKO neurons rescued with Munc13–1 WT and mutants as indicated above. (**B**) Plot of AP-evoked EPSC amplitudes of Munc13–1 disrupting $C_1C_2B$MUN domain interfaces mutants normalized to corresponding Munc13–1 WT. (**C**) Representative traces of synaptic responses induced by 500 mM sucrose from Munc13-1/2

*Figure 4 continued on next page*

*Figure 4 continued*

DKO neurons rescued with the respective Munc13–1 WT and mutants indicated above. (D) Plot of RRP charge of Munc13–1 mutants normalized to corresponding Munc13–1 WT data. (E) Plot of $p_{vr}$ for WT mutant Munc13–1s. (F) Plot showing the average paired-pulse ratios calculated from 2 AP-evoked EPSC amplitudes with a interstimulus interval of 25 ms. (G) Correlation between $p_{vr}$ and paired-pulse ratios from DKO neurons rescued with Munc13–1 WT and mutants. Numbers in plots are *n* values for each group. Error bars represent SEM. Significance and p values were determined by comparison with the corresponding WT using the unpaired Student's *t* test: Mann-Whitney. *p<0.05.

mutant (*Figure 4E*). Remarkably, the R750E,K752E mutation in the $C_1$-$C_2B$ interface caused an even more pronounced decrease in PDBu potentiation (*Figure 6B*), which contrasts with the limited effect of this mutation on vesicular release probability (*Figure 4E*) but correlates with the strong depression observed in the 10 Hz AP trains (*Figure 5C*). These findings strongly support the notion that the interactions between the $C_1$ and $C_2B$ domain observed in our crystal structure are critical for the interplay between the two domains in $Ca^{2+}$- and DAG-dependent presynaptic plasticity.

Because we observed changes in basal release probability and/or RRP size in the three mutants, we next examined how these release parameters are modulated during short-term plasticity induced by action potential trains. We recorded EPSC and sucrose-evoked responses before and 2 s after a 10 Hz action potential train to determine which of those parameters are individually affected (*Figure 7*). For rescues with WT Munc13–1 and the three mutants, the RRP size after the train tended to decrease compared to the RRP size before the train, but the difference was not significant

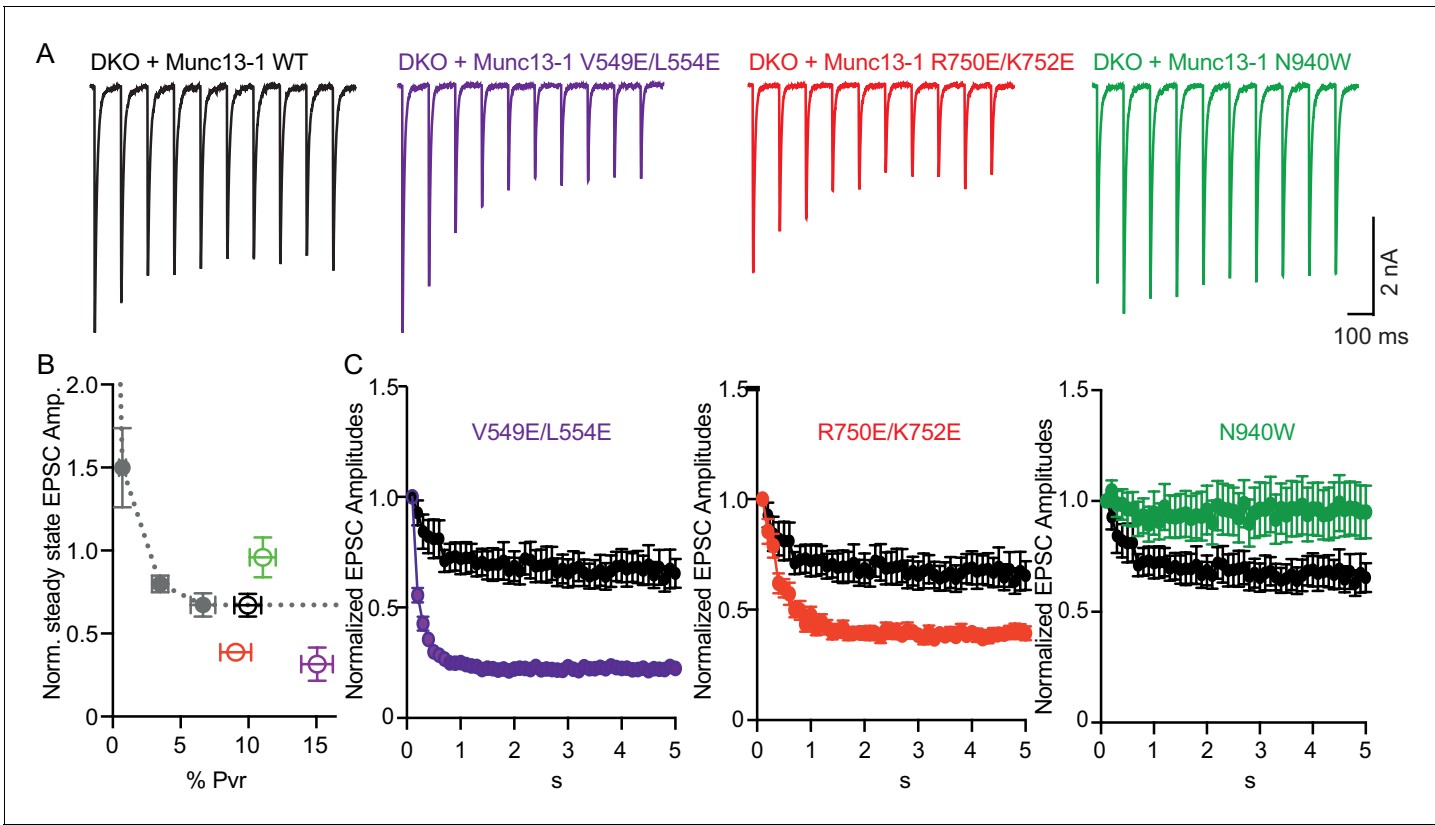

**Figure 5.** Difference in sustained release during a high-frequency action potential train upon disruption of $C_1C_2BMUN$ domain interfaces. (A) Exemplary traces of EPSCs evoked by 10 Hz stimulation trains of Munc13-1/2 DKO neurons rescued with WT and mutant Munc13–1s. (B) Correlation between the amount of steady-state EPSC amplitudes at the end of the 10 Hz train and vesicular release probability. Dotted gray curve provides the correlation of steady-state depression and the release probability by applying AP trains at various external $Ca^{2+}$ concentrations in WT neurons. (C) Analysis of 50 EPSC amplitudes evoked at 10 Hz, which were normalized to the first EPSCs and plotted over time. WT control (black circles) is identical in all three groups.

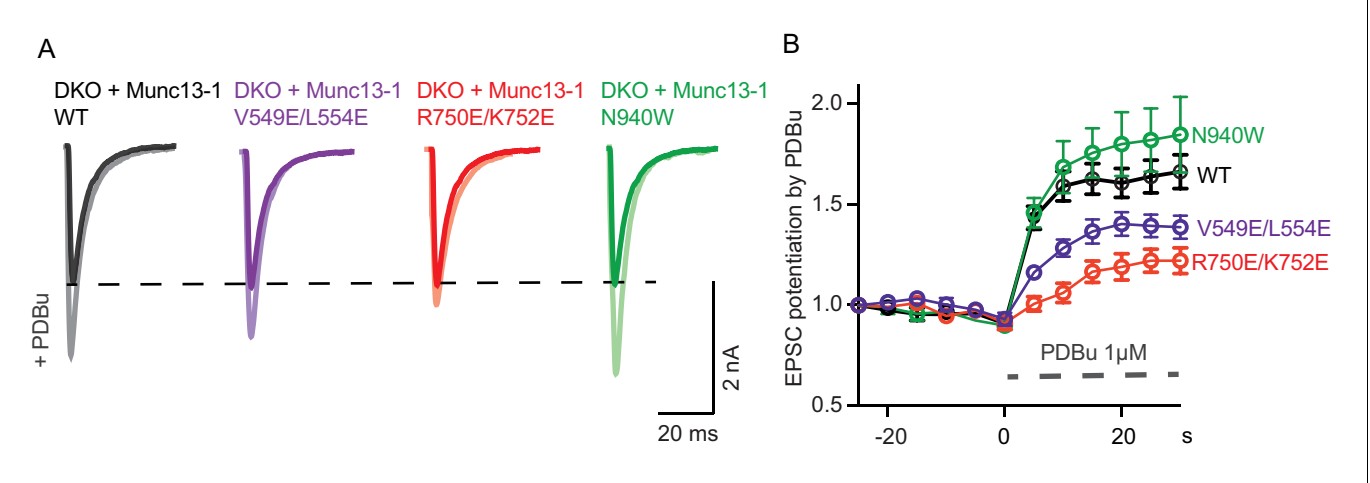

**Figure 6.** Effect of mutations that disrupt the C1C2BMUN domain interfaces on the potentiation of release caused by the activation of the C1 domain by phorbol ester. (**A**) Exemplary EPSC traces (dark colors) from WT and mutant groups and their corresponding EPSCs after PDBu application (light colors). (**B**) Potentiation of EPSC amplitudes by 1 μM PDBu evoked at 0.2 Hz in Munc13-1/2 DKO neurons expressing WT or mutants. The relative PDBu potentiation was calculated by normalizing the EPSC amplitude in PDBu with the presiding EPSC recorded in control extracellular solution. The solid symbols represent the normalized EPSC values in each time point (± SEM).

(*Figure 7B*). On the other hand, the EPSC amplitude increased for the WT rescue after the train (*Figure 7D,F*), thus leading to an increased release probability with respect to the Pvr observed before the train (*Figure 7E*). These results indicate that in the WT rescues the train does not have a substantial effect on the RRP, but the efficiency of evoked release increases due to some alteration (s) of the release machinery.

In rescues with the V549E,L554E mutation in helix H1, comparison of RRP size and release probability before and after a 10 Hz train showed a failure to potentiate the RRP size, as observed in the WT rescue, but also a failure to increase the vesicular release probability (*Figure 7B,E*). These results contrast with the impairment of vesicle priming and increment of vesicular release probability observed for the V549E,L554E mutant in naïve synapses (*Figure 4D,E*). Thus, the enhanced depression during the AP train observed for this mutant compared to WT Munc13–1 (*Figure 5C*) was due to a failure to potentiate release probability and not due to a failure in vesicle replenishment. Similar results were obtained for the rescue with Munc13–1 bearing the R750E,K752E mutation in the $C_1$-$C_2$B interface (*Figure 7B,D–F*), which in naïve synapses caused no impairment of RRP size and vesicular release probability (*Figure 4D,E*), but an enhanced depression of EPSC responses during AP trains (*Figure 5C*). Thus, this enhanced depression was not due to a failure in vesicle replenishment but to a failure to potentiate release probability, as observed for the V549E,L554E mutant. Finally, the N940W mutation in the $C_2$B-MUN interface, which leads to a 40% reduction in RRP size (*Figure 4D*) but no change in release probability (*Figure 4E*) in naive synapses, caused a selective increase in vesicular release probability after the AP train, similar to what is seen in WT rescues but with larger degree of potentiation (*Figure 7B,D–F*). Thus, the finding that this mutant exhibits much less depression than WT during the AP train (*Figure 5C*) arises because this mutant is more likely to potentiate release probability during AP trains than WT Munc13–1.

Overall, our electrophysiological data provide compelling evidence for functional and structural interactions among the domains of the Munc13–1 $C_1C_2$BMUN fragment. These interactions play distinct roles in regulating vesicle priming, vesicle release probability and activity-dependent changes in release probability, thus having different impact on the ability of synapses to dynamically respond to incoming AP patterns.

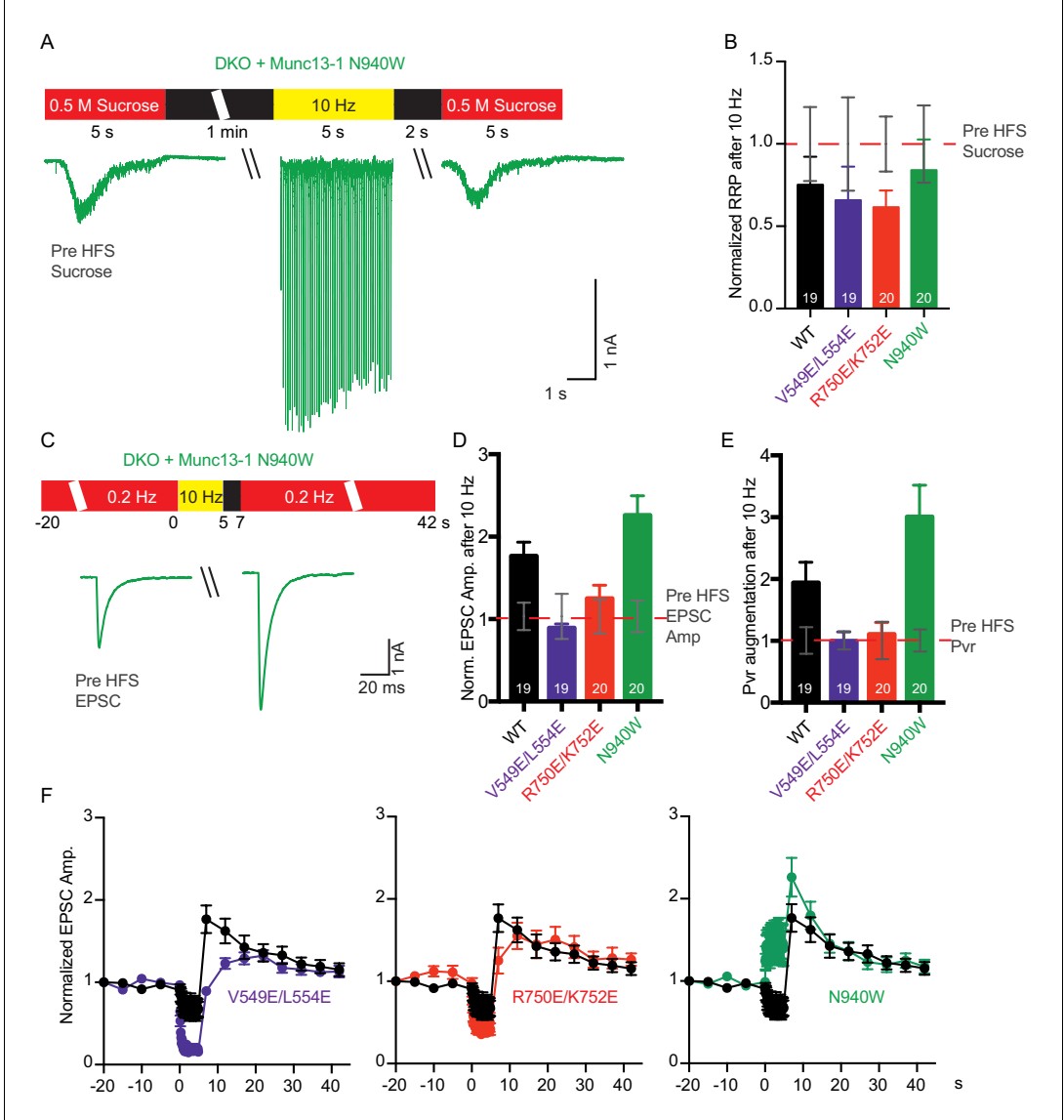

**Figure 7.** Effect of mutations that disrupt the C1C2BMUN domain interfaces on the potentiation of release caused by high-frequency stimulation. (**A**) Schematic diagram illustrating experimental design and example traces of synaptic responses induced by 500 mM sucrose from Munc13-1/2 DKO rescued with Munc13–1 N940W mutant before and after a 10 Hz action potential train. (**B**) Normalized summary plot of RRP charge of Munc13-1/2 DKO neurons rescued with the respective Munc13–1 WT and mutants indicated below. The RRP charges from the different groups were normalized to corresponding RRP charge recorded 1 min before the 10 Hz train (dashed red line). (**C**) Schematic diagram illustrating experimental design and example traces of EPSCs from Munc13-1/2 DKO rescued with Munc13–1 N940W mutant before and after a 10 Hz action potential train. (**D**) Normalized plot of AP-evoked EPSC amplitudes of Munc13-1/2 DKO neurons rescued with the respective Munc13–1 WT and mutants indicated below. The EPSC amplitudes were normalized to the corresponding EPSC recorded before the 10 Hz train (doted red line). (**E**) Plot of the estimated Pvr of Munc13-1/2 DKO neurons rescued with the respective Munc13–1 WT and mutants indicated below normalized to the corresponding Pvr before high-frequency stimulation (preHFS) (doted red line). (**F**) Normalized EPSC amplitudes of Munc13-1/2 DKO neurons rescued with the respective Munc13–1 WT and mutants indicated in each graph, in response to a low-frequency stimulus train (0.2 Hz) that is interrupted by a 5 s 10 Hz stimulus train, as outlined in panel (**C**). Numbers in plots are n values for each group. Error bars represent SEM.

## Effects of disruption of $C_1C_2$BMUN domain interfaces on reconstituted liposome fusion assays

Our reconstitution experiments, which monitor fusion between synaptobrevin liposomes (V-liposomes) and syntaxin-1-SNAP-25 liposomes (T-liposomes) in the presence of Munc18–1, NSF, αSNAP

and a Munc13–1 fragment spanning its $C_1$, $C_2B$, MUN and $C_2C$ domains ($C_1C_2BMUNC_2C$), recapitulate multiple features of neurotransmitter release, in particular the absolute requirement of Munc18–1 and Munc13–1 for membrane fusion (*Liu et al., 2016*). Hence, we used these experiments to examine the effects of disrupting Munc13–1 domain interfaces on membrane fusion in vitro and compare them to those observed in our electrophysiological studies. Unfortunately, multiple attempts to express $C_1C_2BMUNC_2C$ bearing the V549E,L554E mutation failed to yield sufficient amounts of soluble protein, and we were also unable to obtain $C_1C_2BMUNC_2C$ bearing a single V549E or L554E mutation, suggesting that both substitutions strongly destabilize the fragment. Since the V549E,L554E mutant was expressed at levels comparable to WT Munc13–1 in our rescue experiments, it is plausible that this mutation is less destabilizing in the context of these experiments because of interactions of that region with N-terminal sequences of Munc13–1, with other components of the release machinery, or with molecular chaperones.

In our reconstitution experiments, we use an assay that simultaneously measures lipid mixing and content mixing (*Zucchi and Zick, 2011*; *Liu et al., 2016*) to ensure that true membrane fusion is observed, as lipid mixing can occur without content mixing [e.g. (*Chan et al., 2009*; *Zick and Wickner, 2014*)]. We start the experiments in the absence of $Ca^{2+}$, and $Ca^{2+}$ is added after 300 s to examine its effect on liposome fusion. Under the standard conditions of our experiments, in which we use 500 nM $C_1C_2BMUNC_2C$ and we include DAG and $PIP_2$ in the T-liposomes, fusion is highly efficient upon $Ca^{2+}$ addition and the $Ca^{2+}$ dependence arises from $Ca^{2+}$ binding to the $C_2B$ domain (*Liu et al., 2016*). In initial experiments, we observed only mild effects of the R750E,K752E and N940W mutations in Munc13–1 $C_1C_2BMUNC_2C$ on fusion. To allow better discrimination, we lowered the $C_1C_2BMUNC_2C$ concentration to 100 nM, and we compared results obtained with T-liposomes that contained DAG+$PIP_2$, DAG only, or $PIP_2$ only (*Figure 8*). WT $C_1C_2BMUNC_2C$ was still highly active under all these conditions, supporting $Ca^{2+}$-dependent membrane fusion that strictly required Munc13–1 $C_1C_2BMUNC_2C$ and was optimal when the T-liposomes included both DAG and $PIP_2$ (*Figure 8* and *Figure 8—figure supplement 1*). When the T-liposomes contained DAG and $PIP_2$, the N940W mutation in the $C_2B$-MUN interface slightly impaired fusion while the R750E,K752E mutation in the $C_1$-$C_2B$ interface led to a considerably stronger impairment (*Figure 8A,B*; see *Figure 8—figure supplement 2* for quantification). This impairment became more overt with T-liposomes containing only DAG or only $PIP_2$, while the effects of the N940W mutation remained mild (*Figure 8C–F*).

To test whether the effects of the R750E,K752E mutation in these experiments are indeed due to disruption of the $C_1$-$C_2B$ interface, we made a 'reversal' quadruple mutant where we mutated two of the acidic residues from the $C_1$ domain in its interface with the $C_2B$ domain (*Figure 1E*) to basic residues (R750E,K752E,E582R,E584K). These two additional substitutions partially compensated for the effects of the R750E,K752E mutation, as the quadruple mutation recovered some of the activity lost in the R750E,K752E mutant (*Figure 8*). The recovery was modest, but this result is not unexpected because the orientation of the mutated side chains in the quadruple mutant may not be as favorable to form salt bridges as in the WT protein. Hence, the observed functional recovery in the quadruple mutant supports the notion that the effects of the R750E,K752E mutation in the fusion assays are due to disruption of $C_1$-$C_2B$ interactions. We note that all these reconstitution experiments were performed in the absence of synaptotagmin-1 because this $Ca^{2+}$ sensor does not have a marked influence in the results obtained in these bulk assays under the conditions used, although it may cause an acceleration of $Ca^{2+}$-dependent fusion that can only be detected at faster time scales than those used in our measurements (*Liu et al., 2016*). Correspondingly, the relative activities of the WT and mutant $C_1C_2BMUNC_2C$ Munc13–1 fragments in reconstitution experiments that incorporated synaptotagmin-1 into the V-liposomes yielded similar results (*Figure 8—figure supplement 3*) to those obtained in the absence of synaptotagmin-1 (*Figure 8A,B*).

The impairment of fusion caused by the R750E,K752E in our reconstitution assays correlates with the impairment of evoked release observed in the rescue experiments, as well as with the strong depression observed for this mutation in the 10 Hz trains and the impairment of PDBu-induced augmentation (*Figures 4–6*). Altogether, our results strongly support the notion that the $C_1$-$C_2B$ interface observed in the crystal structure of $C_1C_2BMUN$ is important for evoked release and for cooperation between the $C_1$ and $C_2B$ domains in membrane binding during repetitive stimulation to generate an activated state that is stabilized by binding of the $C_1$ domain to DAG and of the $C_2B$ domain to $Ca^{2+}$-$PIP_2$. Thus, as we noted previously (*Liu et al., 2016*), our reconstitutions including

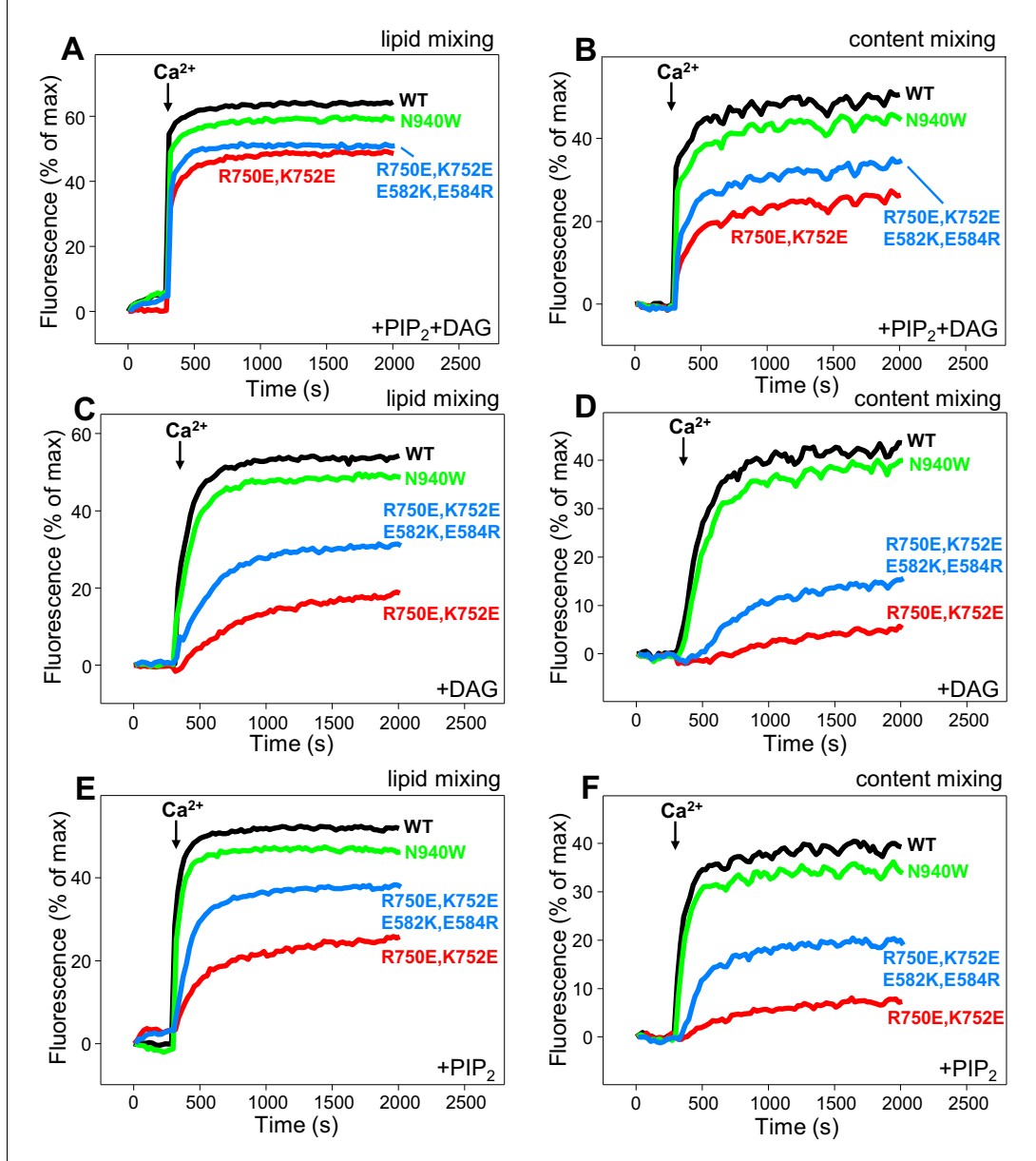

**Figure 8.** Effects of mutations that disrupt $C_1C_2BMUN$ interfaces on membrane fusion in reconstitution assays. Lipid mixing (**A,C,E**) between V- and T-liposomes was measured from the fluorescence de-quenching of Marina Blue-labeled lipids and content mixing (**B,D,F**) was monitored from the development of FRET between PhycoE-Biotin trapped in the T-liposomes and Cy5-Streptavidin trapped in the V-liposomes. The assays were performed in the presence of Munc18–1, NSF-αSNAP and WT or mutant Munc13–1 $C_1C_2BMUNC_2C$ fragments as indicated. Experiments were started in the presence of 100 μM EGTA and 5 μM streptavidin, and $Ca^{2+}$ (600 μM) was added after 300 s.

The following figure supplements are available for figure 8:

**Figure supplement 1.** Dependence of liposome fusion on Munc13–1 $C_1C_2BMUNC_2C$, DAG and $PIP_2$.

**Figure supplement 2.** Quantification of the lipid and content mixing experiments of *Figure 8*.

**Figure supplement 3.** Effects of mutations that disrupt $C_1C_2BMUN$ interfaces on membrane fusion in reconstitution assays including synaptotagmin-1.

DAG and PIP$_2$ in the T-liposomes may recapitulate more closely this activated state than the normal state that leads to release evoked by a single action potential. Interpreting the results of the N940W mutation in the reconstitutions is hindered by the fact that this mutation impaired evoked and sucrose-induced release but had less depression than WT in the 10 Hz trains and a slightly higher potentiation by PDBu (*Figures 4–6*). The small impairment of fusion observed in the reconstitutions for this mutant may reflect a balance between the inhibitory and stimulatory effects observed in the rescue experiments but, overall, the mild nature of the effect on fusion appears to be more reminiscent of the small (albeit contrary) effects observed in the PDBu treatment, thus supporting also the notion that the reconstitutions emulate an activated state of the release machinery.

## Discussion

Munc13–1 acts as a master regulator of neurotransmitter release, playing a crucial role in release itself and multiple functions in integrating the effects of diverse signals that alter release probability during presynaptic plasticity. Thus, elucidating how the functions of the multiple domains of Munc13–1 are coordinated to control release and plasticity constitutes a major, fundamental challenge in neuroscience. The crystal structure of Munc13–1 C$_1$C$_2$BMUN and the functional data presented here represent critical steps to meet this challenge, revealing a highly elongated structure that is key to understanding how Munc13–1 bridges the vesicle and plasma membranes to control SNARE complex formation, and how the C$_1$ and C$_2$B domains mediate presynaptic plasticity processes that depend on Ca$^{2+}$ and DAG.

Notable features of the structure of C$_1$C$_2$BMUN include its length, which is close to 20 nm and is thus comparable to the radius of a synaptic vesicle, and the interfaces between its different domains. The observation that the three mutations designed to disrupt domain interfaces observed in the structure have distinct effects on evoked release, vesicle priming, vesicle release probability, high-frequency stimulation and PDBu-induced facilitation demonstrates the functional relevance of the overall structure and shows that the different domain interfaces perturbed by the mutations play differential roles in release and short-term presynaptic plasticity. This conclusion is further supported by the observation that each of the three mutations uniquely alters the normal relation between vesicle release probability and steady-state EPSC amplitude in 10 Hz AP trains in neurons expressing WT Munc13–1 (*Figure 5B*), a relation that arises from the natural increase in RRP depletion rate as vesicle release probability increases. Similarly, the three mutations also change the normal inverse relation between the extent of PDBu-induced facilitation and the vesicle release probability. This finding is not surprising, as the effects of PDBu are likely to be at least partially related to those observed in 10 Hz trains. Thus, phorbol ester activation of the Munc13 C$_1$ domain is believed to mimic the effects of increased DAG levels during repetitive stimulation, which result from accumulation of intracellular Ca$^{2+}$ and activation of PLCs (*Rhee et al., 2002*). Accumulation of Ca$^{2+}$ also activates Munc13s by binding to the C$_2$B domain (*Shin et al., 2010*).

The finding that the DAG/phorbol ester-binding region of the C$_1$ domain and the Ca$^{2+}$-binding loops of the C$_2$B domain are close to each other and point in the same direction in our C$_1$C$_2$BMUN structure (*Figure 2*) suggest that Ca$^{2+}$, DAG and PIP$_2$ can cooperate in inducing binding of Munc13–1 to the plasma membrane, indicating that these different forms of activation are closely interrelated. These conclusions are supported by the physiological effects of the R750E,K752E mutation in the C$_1$-C$_2$B interface. This mutation leads to a modest impairment of evoked release and to no significant change in the RRP, resulting also in no significant change in vesicular release probability (*Figure 4*). However, this mutation causes a considerably stronger depression during 10 Hz trains than observed for WT Munc13–1 and to severe impairment of PDBu-dependent facilitation (*Figures 5* and *6*). These results suggest that the interaction between the C$_1$ and C$_2$B domains observed in our crystal structure is important for evoked release but is particularly critical for activation of Munc13–1 by phorbol esters as well by Ca$^{2+}$ and DAG during repetitive stimulation. Note also that this mutation strongly disrupts the ability to increase the release probability after repetitive stimulation observed for WT Munc13–1 (*Figure 7E*). Because the C$_1$ and C$_2$B domains pack at the N-terminal end of the long helical structure, far from the middle region of the MUN domain that contains the NF residues involved in opening syntaxin-1 (*Yang et al., 2015*) (*Figure 1B*), our structure does not support models whereby the C$_1$ and C2B domains impair the activity of the MUN domain by direct intramolecular interactions, and binding to DAG or Ca$^{2+}$ releases these inhibitory interactions [e.g.

(*Rizo and Rosenmund, 2008*)]. Instead, our structure suggests that activation of Munc13–1 by PDBu or by repetitive stimulation involves cooperative binding of the $C_1$ and $C_2B$ domains to the plasma membrane in a defined orientation that is promoted by increases in the levels of DAG and intracellular $Ca^{2+}$; this activated state may be more effective in mediating priming and/or downstream events leading to fusion than the state existing under resting conditions, which may involve a different orientation of Munc13–1 with respect to the plasma membrane favored by $Ca^{2+}$- and DAG-independent interactions involving multiple basic residues of the $C_1$ and $C_2B$ domains (*Figure 2C*; see also *Figure 9* and discussion below).

The V549E,L554E mutation designed to disrupt the packing of helix H1 against the linker sequence and the MUN domain decreases the RRP but not the amplitude of the EPSCs induced by a single action potential, resulting in an increased vesicle release probability (*Figure 4*) that mirrors an enhancement in spontaneous release (*Figure 3*). Correspondingly, this mutant exhibits more depression during 10 Hz trains than WT neurons, but the depression is stronger than expected from the observed vesicle release probability (*Figure 5B*). The V549E,L554E mutation also leads to a strong impairment in PDBu-induced facilitation (*Figure 6*) that likely arises from the increased Pvr. Overall, the effects of this mutation resemble those caused by the H567K mutation that is expected to unfold the $C_1$ domain (*Rhee et al., 2002*; *Basu et al., 2007*). The decreases in the RRP observed for both the V549E,L554E and the H567K mutants may result from disruption of coupling between the C-terminal region and the N-terminus containing the $C_2A$ domain, which contributes to the docking-priming activity of Munc13–1 (MC and CR, unpublished results). The increased release probability observed for both mutants suggests that the disruptive structural effects of the two mutations may mimic to some extent the changes involved in activating Munc13–1 during repetitive stimulation, which may involve in part the release of inhibitory effects caused by N-terminal sequences. However, note that the V549E,L554E mutation prevented the increase in Pvr observed after repetitive stimulation for WT Munc13–1 (*Figure 7E*), indicating that different features govern this increase in Pvr and the release probability in naïve synapses.

The N940W mutation in the $C_2B$-MUN interface impairs evoked release and decreases the RRP, leading to a release probability similar to that of WT Munc13–1 (*Figure 4*). However, this mutant exhibits smaller depression during 10 Hz trains than WT Munc13–1 and a slightly larger PDBu-induced facilitation (*Figures 5* and *6*). These results suggest that the packing of the $C_2B$-MUN interface observed in our structure is important for normal vesicle priming and release caused by a single action potential, but the alteration of this interface caused by the N940W mutation favors the activated state that is generated during repetitive stimulation. This view is reinforced by the finding that the N940W mutation causes a larger increase in release probability after repetitive stimulation than that observed with WT Munc13–1 (*Figure 7E*).

The neurotransmitter release machinery is highly complex, including several large proteins of hundreds of kDa that form the active zone in addition to the SNARES, Munc18–1, Munc13–1, NSF, αSNAP, synaptotagmin-1 and multiple additional components (*Südhof, 2013*). Without knowing how Munc13–1 interacts with other proteins, interpretation of the structure of Munc13–1 $C_1C_2BMUN$ in terms of specific models of neurotransmitter release is necessarily speculative. Nevertheless, the structure poses critical constraints on working models of release and its regulation. For instance, the model of *Figure 9A* is based on the finding that the Munc13–1 $C_1C_2BMUNC_2C$ fragment bridges V-liposomes to T-liposomes in the absence of $Ca^{2+}$ (*Liu et al., 2016*). Such bridging might facilitate the activity of the MUN domain in opening syntaxin-1 to initiate SNARE complex formation and may involve interactions of the polybasic face of the $C_1$-$C_2B$ region (*Figure 2C*) with the T-liposomes and of $C_2C$ with the V-liposomes. Binding of the $C_1$ domain to DAG and of the $C_2B$ domain to $Ca^{2+}$ during repetitive stimulation could change the orientation of Munc13–1 with respect to the plasma membrane (*Figure 2B*) in such a way that the two membranes are brought into closer proximity (*Figure 9A*) and the SNARE complex is formed more efficiently. Importantly, if Munc13–1 is located between the two membranes, progress toward complete SNARE complex assembly and fusion would require release of Munc13–1 from the fusion complex because otherwise its large size would hinder further membrane proximity. In this scenario, Munc13–1 could not underlie changes in release probability directly, but its enhanced activity could lead to an increase in the number of preassembled SNARE complexes that might underlie the increases in release probability caused by up-regulation of DAG, by PDBu treatment or by the K630W mutation in a $Ca^{2+}$ binding loop of the Munc13–2 $C_2B$ domain (*Rhee et al., 2002*; *Shin et al., 2010*). However, it is difficult to explain with this type of

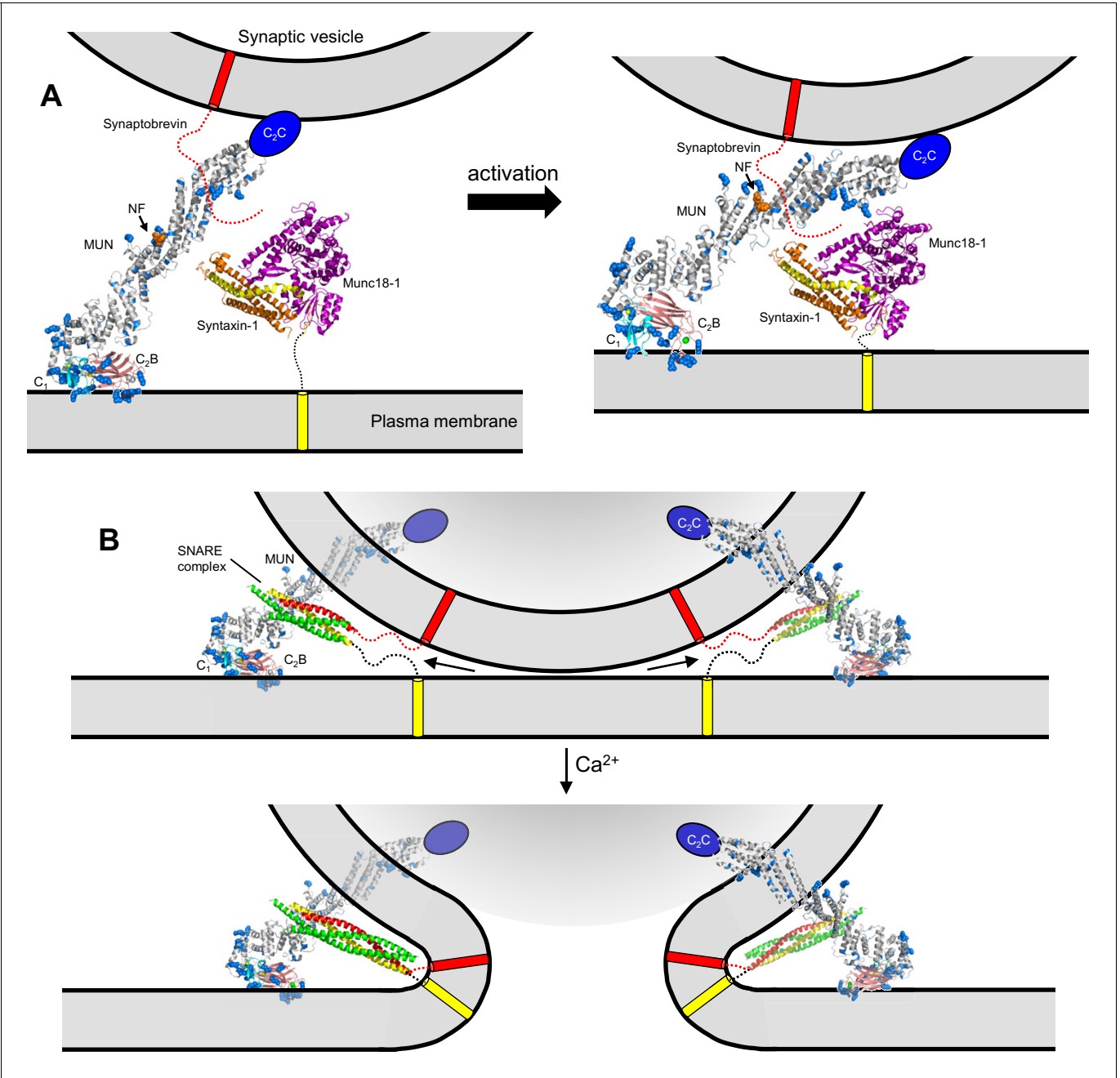

**Figure 9.** Models of neurotransmitter release inspired by the structure of $C_1C_2BMUN$ and our functional data. (**A**) $C_1C_2BMUNC_2C$ is represented by the structure of $C_1C_2BMUN$, with the $C_2B$ domain replaced by the structure of the isolated $Ca^{2+}$-bound $C_2B$ domain (colored in salmon), and by a blue ellipse corresponding to the $C_2C$ domain. $C_1C_2BMUNC_2C$ is shown bridging the two membranes through interactions of the vesicle membrane with the $C_2C$ domain and the plasma membrane with the basic surface formed by the $C_1$ and $C_2B$ domains (left panel) or with the DAG-binding region of the $C_1$ domain and the $Ca^{2+}$-binding region of the $C_2B$ domain (right panel) (see *Liu et al., 2016*). Arginine and lysine side chains are shown as blue spheres. The $C_1$ domain ribbon is in cyan and $Zn^{2+}$ ions are shown as yellow spheres. The two $Ca^{2+}$ ions bound to the $C_2B$ domain are shown as green spheres on the right; on the left, they are shown as gray spheres to represent that the sites are not occupied. The NF sequence involved in opening syntaxin-1 (*Yang et al., 2015*) is represented by orange spheres. The closed syntaxin-1-Munc18–1 complex (PDB code 3C98) is shown to scale to allow comparison of its size with that of $C_1C_2BMUN$. The C-terminus of the syntaxin-1 SNARE motif and the cytoplasmic region of synaptobrevin are shown as dashed curved lines, and their transmembrane regions are represented by cylinders. B. $C_1C_2BMUNC_2C$ is represented as in A but, instead of being located between the two membranes, is bridging the two membranes from a peripheral location. The model must be visualized in three dimensions, such that the $C_2C$ domain is bound to the outer surface of the vesicle membrane rather than inserted into the vesicle lumen. On the right side, the $C_2C$ domain is in the front, whereas in the left side the $C_2C$ domain is in the back of the vesicle, which is represented by a semitransparent gray surface. In both panels, $C_1C_2BMUNC_2C$ is shown in the orientation proposed for the activated state induced during repetitive stimulation, where binding to the

*Figure 9 continued on next page*

*Figure 9 continued*

plasma membrane is mediated by the DAG-binding region of the $C_1$ domain and the $Ca^{2+}$-binding region of the $C_2B$ domain. The SNARE complex (PDB code 1N7S) (*Ernst and Brunger, 2003*), with the SNARE motifs in green for SNAP-25, in red for synaptobrevin and in yellow for syntaxin-1, is shown partially assembled at the top and fully assembled at the bottom. The arrows in the top panel are meant to illustrate that complete assembly of the C-terminus of the SNARE complex might pull the membranes radially outwards, which could create strong membrane tension to trigger membrane fusion. See text for further details.

model why the H567K mutation in the $C_1$ domain does not alter the vesicle refilling rate under resting conditions while increasing the Pvr (*Rhee et al., 2002*), and why PDBu changes the vesicle release probability without altering the RRP (*Basu et al., 2007*).

The distinct effects of Munc13 mutations on evoked release, RRP, vesicular release probability, synaptic responses to repetitive stimulation and PDBu-dependent facilitation [*Figures 3–7* and (*Rhee et al., 2002*; *Basu et al., 2007*; *Shin et al., 2010*) are more readily explained by models postulating that Munc13–1 remains bound to the SNARE complex and/or other components of the release apparatus after SNARE complex assembly, forming part of the fusion complex. Such models predict at least two major actions for Munc13s, one in docking-priming (i.e. orchestrating SNARE complex assembly) and another in fusion. The length of the $C_1C_2BMUN$ structure suggests that the Munc13–1 C-terminal region cannot remain in the central space between two membranes (as in *Figure 9A*) after the SNARE complex is even partially assembled. However, the activities of Munc13–1 in bridging the two membranes and mediating SNARE complex assembly, as well as potential downstream functions, could be performed with Munc13–1 located in the periphery of the membrane-membrane interface, with an arrangement where the large size of Munc13–1 does not hinder completion of SNARE complex assembly and membrane fusion (e.g. *Figure 9B*). With such an arrangement, reorientation of Munc13–1 upon binding to DAG and $Ca^{2+}$-$PIP_2$ could still underlie an increased activity in facilitating SNARE complex formation, but such reorientation could also lead to an optimization of the primed state that can lead more readily to membrane fusion upon $Ca^{2+}$ influx than that observed without Munc13–1 activation. This separation of the roles of Munc13–1 into at least two steps can easily explain the differential effects of the V549E,L554E, R750E,K752E and N940W mutations in the different functional assays, including the finding that N940W impairs priming and evoked release, but enhances Munc13–1 function during 10 Hz trains or PDBu treatment (*Figures 4–7*).

Clearly, the primed fusion complex depicted in *Figure 9B* (top) might contain additional components such as Munc18–1, αSNAP, NSF, synaptotagmin-1 and complexins, among others. An attractive feature of this model is that it places the SNAREs on the periphery of the membrane-membrane interface rather than between the membranes, consistent with electron microscopy images of SNARE-mediated liposome fusion intermediates (*Hernandez et al., 2012*) and of presynaptic terminals showing practically direct contact between docked-primed vesicles and the plasma membrane [e.g. (*Harlow et al., 2001*)]. With this architecture, the SNARE complex could not induce fusion by bringing the vesicle and plasma membranes into proximity, as they are already in contact before $Ca^{2+}$ influx, but could pull the two membranes radially outwards as assembly of the SNARE complex is completed (*Figure 9B*). In this case, fusion would result from the tension created in both membranes, which would favor transient formation of non-bilayer intermediates that might or might not resemble a stalk. Components of the fusion complex that are bound to the SNARE complex and bridge the membranes (e.g. Munc13–1 in *Figure 9B*) would play an important role to establish support points from which the SNAREs can exert their force on the membranes more efficiently.

As discussed in the results section, the strong effect of the R750E,K752E mutation in our fusion assays (*Figure 8*) suggests that our reconstitutions recapitulate better the mechanism of release during repetitive stimulation than release evoked by a single action potential. This proposal is supported by the findings that the tight $Ca^{2+}$ dependence of membrane fusion in our reconstitution assays (*Figure 8*) arises largely from $Ca^{2+}$ binding to the Munc13–1 $C_2B$ domain (*Liu et al., 2016*), and that disrupting the $Ca^{2+}$-binding sites of the Munc13–2 $C_2B$ domain did not impair evoked release in rescue experiments but did impair release during repetitive stimulation (*Shin et al., 2010*). However, it is plausible that functional redundancy with other proteins and/or compensatory effects may have masked an actual role for the Munc13–2 $C_2B$ domain in $Ca^{2+}$ sensing during evoked

release, in cooperation with synaptotagmin-1. A primed state that contains Munc13 as key component bodes well for such a $Ca^{2+}$-sensing role of the $C_2B$ domain, which is also suggested by the increased release probability caused by a K630W mutation in the $Ca^{2+}$-binding loops of the Munc13–2 $C_2B$ domain (*Shin et al., 2010*) and may be reflected in the tight $Ca^{2+}$-dependence of our fusion assays.

Clearly, further research is needed to test all these ideas, and it will be particularly critical to assess whether Munc13–1 is indeed part of the primed fusion complex and, if this is the case, how Munc13–1 binds to the SNAREs and other potential components of the complex. Regardless of these possibilities, the structure of $C_1C_2BMUN$ presented here provides a fundamental framework to design and interpret future studies.

## Materials and methods

### Plasmids and recombinant proteins

To express rat Munc13–1 fragments encoding the $C_1C_2BMUN$ and $C_1C_2BMUNC_2C$ regions (residues 529–1531 and 529–1735, respectively; both constructs have residues 1408–1452 from a flexible loop deleted), the corresponding DNA sequences were originated from full-length rat Munc13–1 (NM_022861, L756W, $\Delta$1415–1437, $\Delta$1533–1551, E1666G). $C_1C_2BMUN$ was cloned into pFAST-BacHTb (EcoRI, HindIII); $C_1C_2BMUNC_2C$ was cloned into pFASTBac(EcoRI, HindIII), which was modified by adding a GST tag and a TEV cleavage site in front of the EcoRI cloning site. The constructs were used to generate a baculovirus using the Bac-to-Bac system (Invitrogen; Waltham, MA). Insect cells (sf9) were infected with the baculovirus, harvested about 72–96 hr post-infection, and re-suspended in lysis buffers $C_1C_2BMUN$ (50 mM Tris pH8.0, 250 mM NaCl, 10 mM imidazole); $C_1C_2BMUNC_2C$ (50 mM Tris pH8.0, 250 mM NaCl, 1 mM TCEP). Cells were lysed and centrifuged at 18,000 rpm for 45 min. The clear supernatant of $C_1C_2BMUN$ was incubated with Ni-NTA resin at 4°C for 2 hr, then the beads were washed with: (i) lysis buffer; (ii) lysis buffer containing 1% Triton X-100; (iii) lysis buffer containing 1M NaCl; and (iv) lysis buffer. The protein was eluted in lysis buffer with 150 mM imidazole, incubated with TEV to remove the His-tag, and purified by ion exchange chromatography. The clear supernatant of $C_1C_2BMUNC_2C$ was incubated with GST agarose at room temperature for 2 hr. The beads were washed with: (i) lysis buffer; (ii) lysis buffer containing 1% Triton X-100; (iii) lysis buffer containing 1M NaCl; and (iv) lysis buffer. The protein was treated with TEV protease on the GST agarose at 22°C for 2 hr. Both $C_1C_2BMUN$ and $C_1C_2BMUNC_2C$ were further purified via gel filtration chromatography and were concentrated to 3–4 mg/ml for storage in 10 mM Tris buffer (pH 8.0) containing 10% glycerol, 5 mM TCEP and 250 mM NaCl. The $C_1C_2BMUNC_2C$ mutants were generated by site-directed mutagenesis, and purified as the WT fragment. The finding that the mutants eluted at the same volumes as WT $C_1C_2BMUNC_2C$ in gel filtration and did not exhibit a higher tendency to degradation strongly suggest that the mutations do not cause folding problems.

### Crystallization and X-ray diffraction data collection

Rat Munc13–1 $C_1C_2BMUN$ (529-1407, 1453-1531) in 0.01 M Tris (pH 8.0), 0.25 M NaCl, 10% (v/v) glycerol and 5 mM TCEP was concentrated to 4–6 mg/ml for crystallization using the sitting drop vapor diffusion method. Drops in a ratio of 1 μl protein to 1 μl well solution were equilibrated against 200 μl 0.1 M Tris (pH 8.0–8.5), 0.2 M LiCl, 12% (v/v) PEG 10,000 at 20°C. Multiple crystals appeared spontaneously in 3 days and were used for dilution microseeding. Drops in a ratio of 1 μl protein to 1–2 μl well solution premixed with crystal seeds were equilibrated against 200 μl 0.1 M Tris (pH 8.0–8.5), 0.2 M LiCl, 10–12% (v/v) PEG 10,000 at 20°C, and crystals were harvested within 10 days. Tantalum derivatized crystals were obtained by overnight incubation with solid $Na_2Ta_6Br_{12}$ in mother liquor drops that contained pre-grown single $C_1C_2BMUN$ crystals. Crystals were cryoprotected by successive transfer in increasing steps of 5% ethylene glycol to a final solution of 20–25% (v/v) ethylene glycol, 0.1 M Tris (pH 8.0), 0.2 M LiCl, 0.15 M NaCl, 10%(v/v) glycerol, 5 mM TCEP and flash-cooled in liquid nitrogen.

Native $C_1C_2BMUN$ crystals exhibited the symmetry of space group C2 with unit-cell parameters of a = 176.1 Å, b = 86.4 Å, c = 202.1 Å and $\beta$ = 115.5° and contained two molecules of $C_1C_2BMUN$ per asymmetric unit. $C_1C_2BMUN$ crystals displayed strong anisotropy and were highly non-

isomorphous. Native $C_1C_2BMUN$ crystals diffracted to a $d_{min}$ of 3.35 Å when exposed to synchrotron radiation. All diffraction data were collected at beamline 19-ID (SBC-CAT) at the Advanced Photon Source (Argonne National Laboratory, Argonne, IL, USA) and processed with *HKL3000* (*Minor et al., 2006*), with applied corrections for effects resulting from absorption in a crystal and for radiation damage (*Borek et al., 2003*; *Otwinowski et al., 2003*), the calculation of an optimal error model, and corrections to compensate the phasing signal for a radiation-induced increase of non-isomorphism within the crystal (*Borek et al., 2010*, *2013*). These corrections were crucial for successful phasing. Initial phases for $C_1C_2BMUN$ were obtained by molecular replacement (MR) with *Phaser* (*McCoy et al., 2007*) using the crystal structure of the N-terminally truncated MUN BCD domain (PDB code: 4Y21) (*Yang et al., 2015*) with the coordinates for several loops and residues at the N- and C-termini removed (residues 942–947, 1038–1041, 1193–1196, 1342–1353, 1404–1409, 1452–1469, 1515–1523) as the search model. One C2B domain was located by MR using the crystal structure of the calcium-free C2B domain (PDB code: 3 KWT) as a search model (*Shin et al., 2010*). Phasing using MR versus the native dataset stalled at this point, and an iterative MR-SAD/rigid body refinement procedure was then adopted. MR-SAD phases calculated in *Phaser* from a tantalum bromide dataset collected at the tantalum LIII edge revealed interpretable density for missing helices H7-H9, which were modeled initially with a polyalanine sequence and rigid body refined in *Phenix* (*Adams et al., 2010*). The updated model was placed in the native cell via MR in *Phaser* and rigid-body refined in *Phenix*. Density modification using two-fold non-crystallographic symmetry was performed in *Parrot* (*Cowtan, 2010*), and automated model rebuilding of only the newly added four helices of each MUN domain was performed in *Buccaneer* (*Cowtan, 2006*). Multiple cycles of alternating this procedure between the tantalum and native datasets allowed the modeling and sequence assignment to helices H1-H9, and the placement of the second C2B domain by MR. Subsequently, both C1 domains were located by MR using the NMR structure (PDB code: 1Y8F) as a search model (*Shen et al., 2005*), and the locations were confirmed by an anomalous difference map calculated from a dataset collected at the zinc K-edge (*Figure 1—figure supplement 1*). Iterative model building and refinement were performed with *COOT* and *Phenix*, respectively (*Emsley and Cowtan, 2004*).

Restraints used in the initial cycles of model refinement included non-crystallographic symmetry, secondary structure and reference models for the C1, C2B and MUN domains. As the coordinates for the MUN domain deposited in the PDB (ID 4Y21) had incorrect sequence numbering and exhibited a high level of side chain outliers, a high clashscore and high overall score in *MolProbity* (*Chen et al., 2010*), the coordinates for the MUN domain were re-refined versus the deposited structure factors and the sequence numbering was corrected prior to use as a reference model for restrained refinement of the C1C2BMUN model. The reference model restraints were removed for the final cycles of refinement. A superposition of chains A and B yield a root mean square deviation (r.m.s.d.) of 1.51 Å for 809 aligned Cα carbons (*Figure 1—figure supplement 2*). The final model for C1C2BMUN ($R_{work}$ = 25.4%, $R_{free}$ = 29.0%) contained 1685 residues in two monomers, 4 $Zn^{2+}$ and 2 $Cl^-$ ions. The higher-than-average $R_{free}$ value is probably due to the relative dearth of lattice contacts (*Figure 1—figure supplement 3*) for the CD subdomains of MUN chain A of C1C2BMUN, as evidenced by weak electron density and high average thermal displacement factors (ADP) (151.2 $Å^2$) for those subdomains (residues 1255–1517) (*Figure 1—figure supplement 4*). Due to the high ADP values for the CD subdomain of chain A, the authors recommend that interpretation of the CD subdomain should be performed on residues in chain B. The density for the remaining domains of chain A as well as chain B of C1C2BMUN is strong and well connected; in fact, the average ADPs for residues 541–950 are lower for chain A (41.0 $Å^2$) than chain B (62.4 $Å^2$). Omit maps for regions where site directed mutations were made are shown in *Figure 1—figure supplement 5*. A Ramachandran plot generated with *MolProbity* (*Chen et al., 2010*) indicated that 92.6% of all protein residues are in the most favored regions and 1.2% in disallowed regions. The majority of the outliers in the Ramachandran plot are located in surface loops with weak electron density that connect domains or secondary structural elements. Data collection and structure refinement statistics are summarized in *Table 1*. The coordinates of the $C_1C_2BMUN$ structure and of the refined MUN domain structure have been deposited in the Protein Data Bank with accession numbers 5UE8 and 5UF7, respectively.

## Lentiviral constructs

The cDNAs of Munc13–1 full length and Munc13–1 V549E,L554E, Munc13–1 R750E,K752E and Munc13–1 N940W were generated from rat Munc13–1 (*Basu et al., 2005*) by PCR amplification. The reverse primer harbors a 3xFLAG sequence (Sigma-Aldrich, Hamburg, Germany) to allow expression analysis. The corresponding PCR products were fused to a P2A linker (*Kim et al., 2011*) after a nuclear localized GFP sequence into the lentiviral shuttle vector, which allows a bicistronic expression of NLS-GFP and the Munc13–1-Flag protein under the control of a human *synapsin-1* promotor. Concentrated lentiviral particles were prepared as described (*Lois et al., 2002*).

## Autaptic hippocampal neuronal cultures and lentiviral infection

Animal welfare committees of Charité Medical University and the Berlin state government Agency for Health and Social Services approved all protocols for animal maintenance and experiments (license no. T 0220/09). Hippocampi were dissected from embryonic day 18.5 Munc13 1/2 DKO mouse and enzymatically treated with 25 units/ml of papain for 45 min at 37°C. After enzyme digestion, hippocampi were mechanically dissociated and the neuron suspension was plated onto astrocytes microislands at a final density of 300 cells cm$^{-2}$. Neurons were infected 24 hr after plating with the lentiviral rescue constructs and incubated at 37°C and 5% $CO_2$ for 13–16 days.

## Electrophysiology

Whole-cell voltage clamp recordings were done at room temperature in 13–16 days in vitro (DIV) autaptic hippocampal Munc13- 1/2 DKO neurons expressing Munc13–1 WT, Munc13–1 V549E, L554E, Munc13–1 R750E,K752E or Munc13–1 N940W. Synaptic currents were monitored using a Multiclamp 700B amplifier (Molecular Devices). The series resistance was compensated by 70% and only cells with series resistances <10 MΩ were analyzed. Data were acquired using Clampex 10 software (Molecular Devices, Sunnyvale, CA) at 10 kHz and filtered using a low-pass Bessel filter at 3 kHz. Borosilicate glass pipettes with a resistance between 2 and 3.5 MΩ were used. Pipettes were filled with internal recording solution contained the following (in mM): 136 KCl, 17.8 HEPES, 1 EGTA, 4.6 $MgCl_2$, 4 $Na_2ATP$, 0.3 $Na_2GTP$, 12 creatine phosphate, and 50 U/ml phosphocreatine kinase; 300 mOsm; pH 7.4. During recordings, neurons were continuously perfused with standard extracellular solution including the following (in mM): 140 NaCl, 2.4 KCl, 10 HEPES, 10 glucose, 2 $CaCl_2$, 4 $MgCl_2$; 300 mOsm; pH 7.4. Spontaneous release was measured by recording miniature EPSCs for 30 s at −70 mV. To detect false-positive events 3 mM of kynurenic acid was applied for an equal amount of time. Action potential-evoked EPSCs were triggered by 2 ms somatic depolarization from −70 to 0 mV. The size of the readily-releasable pool (RRP) was determined by the application with a fast flow system of 500 mM sucrose added to the standard extracellular solution for 5 s. Evoked sucrose responses are characterized by a transient inward current followed by a steady state current. The steady state component represents refilling of primed vesicles and was used to define the baseline. The area under the baseline in the transient curve component was quantified to determine the total charge released by the RRP (*Rosenmund and Stevens, 1996*). The vesicular release probability ($p_{vr}$) was calculated by dividing the EPSC charge by the RRP charge.

The paired-pulse stimulation protocol contained two inductions of an AP at an interval of 25 ms (40 Hz). The paired-pulse ratio was calculated by dividing the amplitude of the second EPSC by the amplitude of the first. To analyze release induced by a high frequency stimulation train, EPSCs were evoked at a frequency of 10 Hz for 5 s. To define the quantitative relationship between Pvr and the steady state EPSC amplitudes during a 10 Hz action potential train, we recorded EPSCs and sucrose evoked responses from wildtype neurons. EPSCs and EPSC trains were recorded in external solutions containing 0.5, 1, 2 and 4 mM calcium and 4 mM magnesium. The hypertonic sucrose solution for all responses contained 2 mM calcium and 4 mM magnesium. The Pvr was computed and correlated with the mean amplitude of the last 10 EPSCs of the 10 Hz AP train. Phorbol ester experiments were monitored in the same cell, EPSC amplitudes were recorded at 0.2 Hz for 30 s in the absence of 1 μM PDBu and the following 30 s in its presence. To dissect how an AP train modulates synaptic output, we examined RRP and release probability separately by probing EPSC amplitude and RRP size 2 s following the 10 Hz train. We then compared the relative changes in RRP and EPSC to the corresponding EPSC and RRP values preceding the AP train. An approximately equal number of cells were recorded from control and experimental groups per day from 3 to 4 consecutive days. The

present dataset was acquired from three separate cultures. To minimize variability between cultures the values of mEPSC, EPSC and RRP from each experimental groups of recording were normalized to the mean value of the WT group for each culture. Data were analyzed offline using Axograph X (Axograph Scientific, Sidney, Australia) and the values were normalized to the WT group. Data summation and statistical analyses were performed using Prism 7 (GraphPad). Significance and p values were determined by comparison of each mutant group with the Munc13–1 WT using the unpaired Student's $t$ test: Mann-Whitney.

## Munc13–1 protein expression level quantification by Western blot

Hippocampal neurons from Munc13-1/2 DKO expressing Munc13–1 full length, Munc13–1 V549E, L554E, Munc13–1 R750E,K752E and Munc13–1 N940W mutants were lysed after 15 DIV at 4°C with RIPA lysis buffer including protease inhibitor cocktail-complete mini (Roche Diagnostics, Berlin, Germany). Equal amounts of proteins from the lysates of the four different groups were mixed with Laemmli sample buffer containing 0.1 M DTT, and boiled 5 min at 99°C. Protein lysates were separated on SDS polyacrylamide gels (4–8%% SDS-PAGE) and transferred to a polyvinyl difluoride (PVDF) membrane. Membranes were blocked for 1 hr with 5% skim milk in TBST and incubated at 4°C over night with primary antibodies: anti-Flag M2 (F1804; Sigma-Aldrich), and anti-Living Colors GFP (632375; Clontech, Mountain View, CA). Secondary antibodies were horseradish peroxidase-conjugated (Jackson ImmunoResearch, West Grove, PA). The immunoreactive proteins were detected by ECL Plus Western Blotting Detection Reagents (GE Healthcare Biosciences, Pittsburgh, PA) in a Fusion FX7 detection system (Vilber Lourmat, Eberhardzell, Germany). Data were collected from three separate Munc13-1/2 DKO cultures and analyzed offline using ImageJ.

## Acknowledgements

We thank Cameron Gundersen for fruitful discussions on models of synaptic vesicle fusion. Results shown in this report are derived from work performed at the Argonne National Laboratory, Structural Biology Center at the Advanced Photon Source. The Argonne National Laboratory is operated by the University of Chicago Argonne, LLC, for the US Department of Energy, Office of Biological and Environmental Research, under contract DE-AC02–06 CH11357. This work was supported by grant I-1304 from the Welch Foundation (to JR), by NIH Research Project Award R35 NS097333 (to JR), which continues research previously supported by NIH grants NS037200 and NS049044 (to JR), and by German Research Council grants SFB958 and SFB665 (to CR and MC).

## Additional information

### Competing interests

CR: Reviewing editor, *eLife*. The other authors declare that no competing interests exist.

### Funding

| Funder | Grant reference number | Author |
| --- | --- | --- |
| Deutsche Forschungsgemeinschaft | SFB958 | Christian Rosenmund |
| Deutsche Forschungsgemeinschaft | SFB665 | Christian Rosenmund |
| National Institutes of Health | R35 NS097333 | Josep Rizo |
| Welch Foundation | I-1304 | Josep Rizo |
| National Institutes of Health | NS037200 | Josep Rizo |
| National Institutes of Health | NS049044 | Josep Rizo |

The funders had no role in study design, data collection and interpretation, or the decision to submit the work for publication.

## Author contributions
JX, Data curation, Formal analysis, Investigation, Writing—review and editing; MC, Conceptualization, Data curation, Formal analysis, Investigation, Methodology, Writing—review and editing; YX, VE, XL, TT, Y-ZP, CM, Data curation, Formal analysis, Investigation; DRT, Conceptualization, Data curation, Formal analysis, Investigation, Methodology, Writing—original draft, Writing—review and editing; CR, Conceptualization, Data curation, Formal analysis, Funding acquisition, Investigation, Methodology, Writing—original draft, Project administration, Writing—review and editing ; JR, Conceptualization, Data curation, Formal analysis, Funding acquisition, Investigation, Methodology, Writing—original draft, Project administration, Writing—review and editing

## Author ORCIDs
Cong Ma, http://orcid.org/0000-0002-7814-0500
Diana R Tomchick, http://orcid.org/0000-0002-7529-4643
Christian Rosenmund, http://orcid.org/0000-0002-3905-2444
Josep Rizo, http://orcid.org/0000-0003-1773-8311

## Ethics
Animal experimentation: Animal welfare committees of Charité Medical University and the Berlin state government Agency for Health and Social Services approved all protocols for animal maintenance and experiments (license no. T 0220/09).

# Additional files

## Major datasets
The following datasets were generated:

| Author(s) | Year | Dataset title | Dataset URL | Database, license, and accessibility information |
| --- | --- | --- | --- | --- |
| Tomchick DR, Rizo J, Xu J | 2016 | C1C2BMUN structure | http://www.rcsb.org/pdb/explore/explore.do?structureId=5UE8 | Publicly available at the Protein Data Bank (accession no: 5UE8) |
| Tomchick DR, Rizo J, Xu J | 2016 | Refined MUN domain structure | http://www.rcsb.org/pdb/explore/explore.do?structureId=5UF7 | Publicly available at the Protein Data Bank (accession no: 5UF7) |

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
