## [Decision Letter]

Thank you for submitting your article "Mechanistic Insights into Neurotransmitter Release and Presynaptic Plasticity from Munc13-1 C1C2BMUN Crystal Structure" for consideration by *eLife*. Your article has been favorably evaluated by Richard Aldrich (Senior Editor) and three reviewers, one of whom is a member of our Board of Reviewing Editors. The following individual involved in review of your submission has agreed to reveal his identity: Matthijs Verhage (Reviewer #3).

The reviewers have discussed the reviews with one another and the Reviewing Editor has drafted this decision to help you prepare a revised submission.

Summary:

In this study, the authors combined crystallography with biochemical and functional assays to investigate the structure and function of a functionally active fragment of Munc13 containing its C1, C2B and MUN domains. Munc13 is essential for exocytosis, and it undergoes multiple interactions with other proteins, particularly SNARE proteins, and the plasma membrane, but its precise mechanism of action is not yet understood. Based on the crystal structure that was obtained, the authors designed different mutations to specifically target the interactions between these domains and study the potential cooperation and synergy between the domains. These mutants were expressed in neurons in which the Mun13 isoforms were deleted, thus allowing for assessing the effect of each of the mutations on the function of the protein, yielding novel insights into the cooperative functions of the different domains.

Generally, all reviewers appreciate the novel structure and agree that it constitutes an important contribution to the field. They also are very supportive of the rescue experiments but there are some concerns that need to be addressed, involving additional experiments.

Essential revisions:

1) The data completeness of native data set of the C1C2BMUN fragment is only ~ 80%, the Ramachandran statistics indicates some outliers (Table 1), and the RSRZ score (provided in the PDB validation report) is not ideal. The model might benefit from checking these outliers and regions of poor fit to the density map. Moreover, representative omit electron density maps should be provided as a supplementary figure.

2) There is a structure of the MUN domain reported in Table 1, but it seems that it is not discussed in the text. Apologies if I may have missed it. I assume this structure of the MUN domain includes the extra residues at the N-terminus that were found to belong to the MUN domain?

3) Is there potential flexibility (e.g., bending) of the fragment? Please provide an overlay of the two NCS mates.

4) All three reviewers appreciate the site-directed mutagenesis experiments but are concerned that mutations may result in folding defects. We realize that a structural characterization of the mutants (although certainly desirable and interesting) goes beyond what can be done, particularly when considering that the purification of these mutant proteins may be difficult. However, we suggest to express these mutants (or at least the domains containing the mutants) in a eukaryotic cell and then check expression of the intact fragments by western blotting. Since unfolded proteins are usually degraded, the presence of non-degraded protein would increase confidence that the expressed proteins do not have major folding defects.

5) The phenotype of the N940W mutation is interpreted as "an easier transition for Munc13-1 to a potentiated state" (subsection “Functional consequences caused by disruption of C_1_C_2_BMUN domain interfaces”, last paragraph). However, the functional consequence of this "potentiated state" is not demonstrated. Is there a higher Pvr or more priming in this state? Therefore, priming and Pvr should tested right after the 10Hz train by hypertonic (500mM) sucrose stimulation.

---

## [Author Response]

Essential revisions:

*1) The data completeness of native data set of the C1C2BMUN fragment is only ~ 80%, the Ramachandran statistics indicates some outliers (Table 1), and the RSRZ score (provided in the PDB validation report) is not ideal. The model might benefit from checking these outliers and regions of poor fit to the density map. Moreover, representative omit electron density maps should be provided as a supplementary figure.*

Figure 1—figure supplement 5 is now provided to show the omit electron density maps in the regions where site directed mutations were made in this study, and additional details on the procedures followed for structure determination are described in Materials and methods (subsection “Crystallization and x-ray diffraction data collection”).

Data completeness less than 100% for the native data set is due to the strong anisotropy of X-ray diffraction of the crystals. The corrections applied in *HKL-3000* for effects resulting from absorption in a crystal and for radiation damage to the diffraction data as described in the methods section of the manuscript also performs *B*-factor sharpening; thus the differences observed between the Wilson *B*-value without sharpening (104.7 Å2) versus with sharpening (49.6 Å2). This *B*-factor sharpening improved the connectivity in the electron density maps and allowed for the tracing of the main chain for many loops that connect secondary structural elements. Unfortunately, in many of these loops the density for the side chains is almost or completely non-existent, and the density that is typically observed in a 3.35 Å resolution map does not often indicate where to place main chain carbonyl oxygens. Even when certain residues are modeled into acceptable regions of the Ramachandran plot, subsequent reciprocal space refinement may move these residues to unfavorable regions. In this structure, all of the Ramachandran outliers are in loop regions that connect secondary structural elements.

Further rounds of refinement of the C1C2BMUN model, which included both non-crystallographic symmetry restraints and carefully optimized TLS atomic displacement parameter refinement has reduced the R(free) to 29.0% and the R(work) to 25.4%, with excellent geometry as evidenced by both the PDB validation report and the results from the *MolProbity* web server, with both the Clashscore and the overall *MolProbity* score in the 100^th^ percentile for the resolution range of 3.35 Å +/- 0.25 Å.

Author response image 1.**DOI:**
http://dx.doi.org/10.7554/eLife.22567.022

Author response image 2.**DOI:**
http://dx.doi.org/10.7554/eLife.22567.023

Regarding the RSR Z-score, the PDB in fact recognizes that there are difficulties in interpreting these scores for low-resolution structures, especially ones “that have been refined using *PHENIX* where hydrogen atoms are involved”. For both the C1C2BMUN and MUN domain structures in this study, explicit riding hydrogen atoms were used during model refinement in the program package *PHENIX* (http://www.wwpdb.org/validation/2016/FAQs#ligand-geometry-difference-with-refinement-packages).

Author response image 3.**DOI:**
http://dx.doi.org/10.7554/eLife.22567.024

What could be the source of these difficulties? Some crystallographers think that it may be due to the nature of the RSR Z-score. Figure 13 shows what the PDB tells us regarding how the RSR Z-score is calculated (see also Kleywegt G.J. et al.(2004) The Uppsala Electron-Density Server. *Acta Cryst.* D 60: 2240-2249. https://doi.org/10.1107/S0907444904013253):

Author response image 4.**DOI:**
http://dx.doi.org/10.7554/eLife.22567.025

In Ian Tickle’s paper “Statistical quality indicators for electron-density maps” (Acta Cryst. (2012) D68, 454–467), he addresses the problems with the Real-space R (RSR) as a validation statistic (see p. 456 and section 4.2 of this article). Basically, the values of the limiting radii used for calculation of the atomic scattering factors are chosen arbitrarily and differ between software implementations used to calculate the electron density maps, which causes RSR to vary according to the software used. Additionally, the value of the RSR increases as a function of the increasing value of the *B*-factor. This latter problem with the RSR Z-score relates directly to the C1C2BMUN structure, and most especially to the portions of the structure with high values for the *B*-factor. As mentioned in the Methods section of our paper, the average *B*-factor of residues 1255 – 1517 of C1C2BMUN chain A (151.2 Å2) is considerably higher than the overall average of the rest of the chain (54.1 Å2 for residues 541-1254). Average values do not tell the whole story in this case, but a plot of *B*-factor values versus residues for the two chains are illustrative (the blue trace is for chain A, and the green for chain B). This plot is now provided as Figure 1—figure supplement 4.

The average *B*-factor for chain A rises precipitously at residue 1255, which corresponds with the high RSR Z-score in the PDB validation report (the residues with the red dots).

Author response image 5.**DOI:**
http://dx.doi.org/10.7554/eLife.22567.026

The electron density for this region of chain A is of poor quality; however, there is no indication that this portion of chain A is not present in the crystal used in this experiment (i.e., it is not a fragment due to proteolysis). Two observations support the existence of an intact C1C2BMUN fragment for chain A: 1) SDS PAGE gels of crystalline samples do not indicate proteolysis, and 2) a fragment that terminated at residue 1255, which is in the loop between H19 and H20, would destabilize the fragment by splitting the 5-helical bundle of the C subdomain. Reciprocal-space refinement after removal of residues A1255-A1517 from the model resulted in a worsening of the R(free) and model geometry. It is for this reason that we have included a plot of atomic thermal displacement parameters (*B*-factors) versus residue for both chains as a supplemental figure (Figure 1—figure supplement 4) and included the following statement in the Methods section: “Due to the high ADP values for the CD subdomain of chain A, the authors recommend that interpretation of the CD subdomain should be performed on residues in chain B. The density for the remaining domains of chain A as well as chain B of C1C2BMUN is strong and well connected; in fact, the average ADPs for residues 541-950 are lower for chain A (41.0 Å2) than for chain B (62.4 Å2).”

The high RSR Z-score values in chain B also correspond to localized regions of high average *B*- factor, most commonly in loop regions connecting secondary structural elements.

Author response image 6.**DOI:**
http://dx.doi.org/10.7554/eLife.22567.027

*2) There is a structure of the MUN domain reported in Table 1, but it seems that it is not discussed in the text. Apologies if I may have missed it. I assume this structure of the MUN domain includes the extra residues at the N-terminus that were found to belong to the MUN domain?*

We now clarify in the Results section (subsection “Crystal structure of Munc13-1 C_1_C_2_BMUN”, first paragraph) that this is a re-refinement of the structure (PDB ID 4Y21) reported in Yang X., et al., Nat Struct Mol Biol 22, 547-554 (2015), to correct the residue numbering and improve the geometry prior to use as a reference model for geometric restraints during *PHENIX* refinement of the C1C2BMUN structure. It was deemed useful as a reference model as the MUN domain structure was determined at a higher resolution than the C1C2BMUN structure (2.9 vs. 3.35 Å). Unfortunately the geometry of the coordinates as originally deposited were deemed worse than coordinates relative to a similar resolution in the PDB, especially in terms of Clashscore and Sidechain outliers.

Author response image 7.**DOI:**
http://dx.doi.org/10.7554/eLife.22567.028

A more detailed assessment of the issues with the coordinates can be found by using the *MolProbity* server, which classified the Clashscore in the 91^st^ percentile, and the overall *MolProbity* score in the 75^th^ percentile for the resolution range of 2.90 Å +/- 0.25 Å.

Author response image 8.**DOI:**
http://dx.doi.org/10.7554/eLife.22567.029

After re-refinement of the coordinates versus the original structure factors, the geometric parameters are more reasonable, with both the Clashscore and the overall *MolProbity* score in the 100^th^ percentile for the resolution range of 2.90 Å +/- 0.25 Å.

Author response image 9.**DOI:**
http://dx.doi.org/10.7554/eLife.22567.030

Author response image 10.**DOI:**
http://dx.doi.org/10.7554/eLife.22567.031

The plot of *B*-factor values for the MUN domain versus the two chains of C1C2BMUN are shown in Figure 20 (the blue trace is for C1C2BMUN chain A, the green for C1C2BMUN chain B, and the red for the MUN domain).

Author response image 11.**DOI:**
http://dx.doi.org/10.7554/eLife.22567.032

The high RSRZ values from the PDB validation report for the MUN domain also correspond to localized regions of high average *B*-factor, most commonly in loop regions connecting secondary structural elements.

Author response image 12.**DOI:**
http://dx.doi.org/10.7554/eLife.22567.033

We note that the program *REFMAC* was used in the refinement of the MUN domain for the original PDB deposition of 4Y21.

*3) Is there potential flexibility (e.g., bending) of the fragment? Please provide an overlay of the two NCS mates.*

Figure 1—figure supplement 2 is now provided to show the superposition. In this figure, C_1_C_2_BMUN chain A is green and C_1_C_2_BMUN chain B is red. The magnitude of bending in the fragment is relatively small, and the differences observed in this study are highest in the CD subdomain, which is the region of weakest density in chain A. Note that an alignment of the fragments from residue 541-1254 of chains A and B results in an r.m.s.d. of 0.71 Å, not terribly different than the maximum likelihood coordinate error estimate of 0.41 Å. Please also keep in mind that initial rounds of refinement of the C_1_C_2_BMUN structure used the MUN domain as a reference model for geometry restraints, and the final rounds of refinement removed the reference model restraints but included torsional-angle NCS geometry restraints between chains A and B. For reference we also include below the r.m.s.d. values for the superposition of the MUN domain to each individual C_1_C_2_BMUN chain.

**Monomer #1**
C1C2BMUN chain A**Monomer #2**
C1C2BMUN chain B**#C-αcarbons**
809**r.m.s.d. (Å)**
1.51C1C2BMUN chain A, 541-1254C1C2BMUN chain B, 541-12546400.71C1C2BMUN chain AMUN4650.96C1C2BMUN chain BMUN4791.03

These observations suggest that there is limited flexibility in the overall architecture of C_1_C_2_BMUN, as we now point out in the last paragraph of the subsection “Crystal structure of Munc13-1 C_1_C_2_BMUN”.

*4) All three reviewers appreciate the site-directed mutagenesis experiments but are concerned that mutations may result in folding defects. We realize that a structural characterization of the mutants (although certainly desirable and interesting) goes beyond what can be done, particularly when considering that the purification of these mutant proteins may be difficult. However, we suggest to express these mutants (or at least the domains containing the mutants) in an eukaryotic cell and then check expression of the intact fragments by western blotting. Since unfolded proteins are usually degraded, the presence of non-degraded protein would increase confidence that the expressed proteins do not have major folding defects.*

The rescue experiments already involved expression in eukaryotic cells (neurons) and the Western blots of Figure 3—figure supplement 1 show that full-length proteins were detectable. The mutant C_1_C_2_BMUNC_2_C fragments used for the reconstitution assays were expressed in another type of eukaryotic cell (Sf9). Hence, we believe that expression in yet another type of eukaryotic cell would not be very informative. In the Materials and methods section we have added the following sentence: “The finding that the mutants eluted at the same volumes as WT C_1_C_2_BMUNC_2_C in gel filtration and did not exhibit a higher tendency to degradation strongly suggest that the mutations do not cause folding problems.” Below we show the gel filtration chromatograms obtained for the WT and three C_1_C_2_BMUNC_2_C mutants, but we feel is not necessary to include them into the paper. As we pointed out in the manuscript, we did have problems expressing the V549E,L554E mutant in Sf9 cells. However, the finding that this mutant rescues evoked release as efficiently as WT Munc13-1 (Figure 3) strongly suggests that the mutation does not cause major folding problems.

Author response image 13.**DOI:**
http://dx.doi.org/10.7554/eLife.22567.034

*5) The phenotype of the N940W mutation is interpreted as "an easier transition for Munc13-1 to a potentiated state" (subsection “Functional consequences caused by disruption of C_1_C_2_BMUN domain interfaces”, last paragraph). However, the functional consequence of this "potentiated state" is not demonstrated. Is there a higher Pvr or more priming in this state? Therefore, priming and Pvr should tested right after the 10Hz train by hypertonic (500mM) sucrose stimulation.*

We fully appreciate this criticism, which has led us to analyze which of the release parameters in the rescue mutants were affected during high frequency stimulation. We have performed additional electrophysiological experiments, recording EPSCs and evoked sucrose release immediately after a 10 Hz action potential train to investigate whether RRP or release probability underlie the changes in steady-state responses during the train (data shown in new Figure 7 and described in subsection “Functional consequences caused by disruption of C_1_C_2_BMUN domain interfaces”, seventh and eighth paragraphs). In general, no large change in vesicle priming was observed for WT or any of the mutants, but we found that Pvr was a main modulator underlying train behavior. Thus, the vesicular release probability doubled after the AP train for WT Munc13-1, and increased even more for the N940W mutant. In contrast, no increase in Pvr was observed for the V549E,L554E and R750E,K752E mutants, which depress more strongly than WT Munc13-1 during the AP trains.